**Quantifying Coupling Errors in Atmosphere-Ocean-Sea Ice Models: A Study of Iterative and Non-Iterative Approaches in the EC-Earth AOSCM** 

Valentina Schüller<sup>1</sup>, Florian Lemarié<sup>2</sup>, Philipp Birken<sup>1</sup>, and Eric Blayo<sup>2</sup>

<sup>1</sup>Lund University, Lund, Sweden

<sup>2</sup>Univ. Grenoble Alpes, Inria, CNRS, Grenoble INP, LJK, Grenoble, France

**Correspondence:** Valentina Schüller (valentina.schuller@math.lu.se)

Abstract. The atmosphere, ocean, and sea ice components in Earth system models are coupled via boundary conditions at the sea surface. Standard coupling algorithms correspond to the first step of an iteration, so-called Schwarz waveform relaxation. Not iterating is computationally cheap but introduces a numerical coupling error, which we aim to quantify for the case of a coupled single column model: the EC-Earth AOSCM, which uses the same coupling setup and model physics as its host model, EC-Earth. To this end, we iterate until a reference solution is obtained and compare this with standard, non-iterative algorithms. Understanding the convergence behavior of the iteration, as well as the size of the coupling error, can inform model and algorithm development. Our implementation is based on the OASIS3-MCT coupler and allows to estimate the coupling error of multi-day simulations.

In the absence of sea ice, SWR convergence is robust. Coupling errors for atmospheric variables can be substantial. When sea ice is present, results strongly depend on the model version. In the latest model version, coupling errors in sea ice surface and atmospheric boundary layer temperature are often large. Generally, we find that abrupt transitions between distinct physical regimes in certain parameterizations can lead to substantial coupling errors and even non-convergence of the iteration. We attribute discontinuities in the computation of atmospheric vertical turbulence and sea ice albedo as sources for these problems.

#### 1 Introduction

Earth system models (ESMs) and general circulation models (GCMs) are large, complex computer codes coupling different submodels (components) in time and space. To this end, they exchange (boundary) data, e.g., heat fluxes and temperatures, at regular intervals. As component development progresses and resolution increases, it is expected that aspects of coupling will play a bigger role (Gross et al., 2018). We focus on atmosphere-ocean and atmosphere-ocean-sea ice coupling, where multiple sets of partial differential equations are coupled using boundary conditions. This can be seen as an example of domain decomposition without overlap.

Schwarz waveform relaxation (SWR) methods are iterative coupling algorithms suitable for such problems: if constructed correctly, the coupled problem has a unique solution which the iteration converges to. Standard coupling approaches in state-of-the-art ESMs can be classified as the first iteration of an SWR algorithm. Not iterating is computationally cheap but produces

a numerical coupling error at the air-sea interface. This error is separate from other numerical errors (e.g., those introduced due to non-matching grids in time and space) and from modeling errors such as those resulting from uncertainties in the parameterizations of turbulent air-sea flux components (e.g., Foken, 2006; Large, 2006).

In case of convergence, the SWR method produces a reference solution to quantify the coupling error of standard coupling algorithms in isolation. As opposed to other types of numerical convergence studies, this is possible without violating implicit assumptions of physics parameterizations on time step or grid size (Gross et al., 2018). Specifically, past studies demonstrated that using SWR eliminates phase errors (Marti et al., 2021) and reduces ensemble spread (Connors and Ganis, 2011; Lemarié et al., 2014). The latter suggests a connection between coupling errors and model uncertainties.

Since components are developed independently, it is likely that a given ESM solves an ill-posed problem. By this we mean that choices made with respect to modeling and numerical algorithms result in non-unique solutions and unexpected amplification of small perturbations. The latter aspect has direct implications on model performance, motivating the investigation of such issues. Gross et al. (2018) suggest to verify that the coupling of ESM components is formulated in a robust and consistent manner using SWR; if the iteration does not converge, model development is advised.

We study these aspects in the context of atmosphere-ocean(-sea ice) coupling, where in particular the inclusion of sea ice is a novel contribution: this component has been excluded in past studies of ESM coupling errors. We address the following three research questions: Do iterative coupling methods for a given model converge? If so, how large is the coupling error of state-of-the-art coupling algorithms? If not, which components cause non-convergence?

The interface boundary conditions in atmosphere-ocean-sea ice coupling are part of the vertical physics parameterizations in ESMs and GCMs. It is therefore particularly relevant to study how these parameterizations interact. For this reason, we study SWR algorithms in a coupled single column model (SCM), the EC-Earth coupled atmosphere-ocean single column model (AOSCM, Hartung et al., 2018). SCMs are one-dimensional in space, simulating the physical processes in a vertical column of, in our case, the atmosphere and the ocean. Large scale dynamics are not explicitly modeled but supplied as forcing. Coupled SCMs contain the same coupling physics and numerics as ESMs and GCMs but are cheap to run. They thus bridge a gap between idealized and full complexity models.

The EC-Earth AOSCM uses the same set of physics parameterizations as its host model, EC-Earth (Döscher et al., 2022). It couples the single column versions of the Open Integrated Forecasting System (OpenIFS, atmosphere) and the Nucleus for European Modelling of the Ocean (NEMO, ocean and sea ice) using the OASIS3-MCT coupling software. We make use of three different combinations of component versions: OpenIFS cy40r1 coupled to NEMO 3.6, OpenIFS cy43r3 coupled to NEMO 4.0.1, and OpenIFS cy43r3 coupled to NEMO 4.0.1. The first version corresponds to the EC-Earth 3 AOSCM as described in Hartung et al. (2018). The latter two are development versions on the path to the EC-Earth 4 AOSCM (which will be based on the same components as EC-Earth 4).

We thoroughly investigate the numerical behavior of the EC-Earth AOSCM with respect to the coupling setup. To this end, we have implemented an SWR algorithm based on the OASIS3-MCT coupler, treating OpenIFS and NEMO as black boxes. This allows us to study SWR convergence and compute the coupling error of standard coupling algorithms for multi-day simulations for several near-surface variables. We find that SWR convergence in the EC-Earth AOSCM is robust in ice-free

conditions, allowing us to compute coupling errors. We find that already after two days, temperature coupling errors in the atmospheric boundary layer can reach several degrees in magnitude. The size of these errors seems to be related to the non-smooth mass flux scheme of the vertical turbulence parameterization in OpenIFS. Presence of sea ice in the model versions with NEMO 3.6 consistently leads to very large, unphysical oscillations, indicating issues in the coupled model formulation. In the newest development version, the remaining SWR oscillations are substantially smaller and occur when sea ice starts melting. Our experiments show that these are caused by jumps in the ice albedo parameterization in NEMO 4.0.1. We then test a smoothened transition between melting and drying conditions and find that this resolves these issues. However, substantial coupling errors for atmospheric and ice surface temperature after only two days suggest that further method development for atmosphere-ocean-sea ice coupling is needed.

The remainder of this paper is structured as follows: Section 2 presents the governing equations and boundary conditions at the air-sea interface solved by the EC-Earth AOSCM, both with and without sea ice. Section 3 focuses on coupling algorithms: we present standard approaches, as well as SWR, and explain how we implemented coupling algorithm switching in the EC-Earth AOSCM. In Section 4, we show and discuss numerical results to assess SWR convergence and the coupling error of the EC-Earth AOSCM for two different locations (with and without sea ice). We conclude with a summary of our main findings and their implications.

#### 2 Overview of the EC-Earth AOSCM

In this section, we specify the coupled problem solved by the EC-Earth AOSCM, based on the model description in Hartung et al. (2018). We deliberately disregard terms that are not relevant in the representation of flow near the sea surface. The section begins with the model equations for OpenIFS and NEMO. Afterwards, we give the interface boundary conditions for the atmosphere and ocean models in ice-free conditions, followed by the case of nonzero sea ice cover.

#### 2.1 Atmosphere: OpenIFS

The OpenIFS SCM solves the primitive equations in a vertical column of fluid. The corresponding equations written in pressure coordinates are:

$$\frac{\partial u}{\partial t} = -\omega \frac{\partial u}{\partial p} + F_u + g \frac{\partial \mathcal{J}_u}{\partial p} + f(v - v_g) \tag{1a}$$

$$\frac{\partial v}{\partial t} = -\omega \frac{\partial v}{\partial p} + F_v + g \frac{\partial \mathcal{J}_v}{\partial p} - f(u - u_g)$$
(1b)

$$\frac{\partial T}{\partial t} = -\omega \frac{\partial T}{\partial p} + F_T + g \frac{\partial \mathcal{J}_T}{\partial p} + \frac{RT\omega}{c_p p} + \frac{g}{c_p} \frac{\partial}{\partial p} (\mathcal{F}_{\text{SW}} + \mathcal{F}_{\text{LW}}) + P_T$$
 (1c)

$$\frac{\partial q}{\partial t} = -\omega \frac{\partial q}{\partial p} + F_q + g \frac{\partial \mathcal{J}_q}{\partial p} + P_q \tag{1d}$$

Therein, t and p denote time and pressure, respectively. Equations (1a) and (1b) are the momentum equations for the zonal and meridional wind velocities u and v. Equations (1c) and (1d) are the conservation equations for internal energy and moisture,

with air temperature T and moisture q. In the OpenIFS SCM, the air pressure p is read in from initial conditions and kept constant for the whole simulation. As opposed to the 3D model, no continuity equation is solved. This leaves a closed system of four prognostic variables (u, v, T, q) and four equations.

Here and later in the text,  $\phi$  is a placeholder for any of the prognostic variables in the AOSCM. The first two terms on the right-hand side in all four equations,  $-\omega \partial_p \phi$  and  $F_{\phi}$ , represent vertical and horizontal advection, respectively, with  $\omega$  being the vertical velocity in pressure coordinates. Both advection terms are supplied as forcing, i.e., their values are read in from a file.

The third term in all four equations expresses vertical turbulent transport on subgrid scales, given by the gradient of the vertical turbulent flux

$$\mathcal{J}_{\phi} = \left(-\rho K_{\phi} \frac{\partial \phi}{\partial z} + M(\phi)\right),\tag{2}$$

where  $\rho$  denotes density,  $K_{\phi}$  is the eddy viscosity/diffusivity and  $M(\phi)$  is the so-called convective mass flux to account for the contribution of deep and shallow convection to the turbulent flux. This is referred to as the Eddy-Diffusivity and Mass-Flux (EDMF) approach (Siebesma et al., 2007). At the surface, the mass flux term is assumed to be zero (ECMWF, 2014,  ${\rm SIV}(3.1)$ ). Here, we define the z-coordinate as positive upwards, with the sea surface being located at z=0, and used that  $(1/\rho)\partial_z=-g\partial_p$ .

The last term in the momentum equations combines the Coriolis effect with large-scale pressure gradient forcing, represented by the geostrophic wind velocities  $u_q, v_q$ . Therein, f denotes the Coriolis parameter.

The fourth term in the thermodynamic equation represents the change of internal energy due to work on the volume, with R and  $c_p$  the moist air gas constant and heat capacity of moist air at constant pressure, respectively. This makes use of the equation of state in the atmosphere, which assumes moist air to be an ideal gas.

Radiation leads to heating or cooling of the atmospheric layers and enters the thermodynamic equation via the net short-wave and long-wave radiative fluxes  $\mathcal{F}_{SW}$  and  $\mathcal{F}_{LW}$ . We use  $P_T$  and  $P_q$  in the OpenIFS equations as a placeholder for the impact of clouds on the thermodynamic and moisture equations (ECMWF, 2014, §IV.7). These terms do not contain additional vertical derivatives.

## 2.2 Ocean and sea ice: NEMO and SI<sup>3</sup>


The NEMO model discretizes the oceanic primitive equations (i.e. jointly considering the Boussinesq, incompressible, and hydrostatic assumptions). In the SCM framework, combining the continuity equation and the no-penetration condition through the ocean floor leads to vanishing vertical velocity w(z) = 0 at all depths z. The NEMO SCM equations for the ocean

component are: 115




$$\frac{\partial u}{\partial t} = -\frac{1}{\rho_{\text{ref}}} \frac{\partial}{\partial z} \mathcal{J}_u + fv \tag{3a}$$

$$\frac{\partial v}{\partial t} = -\frac{1}{\rho_{\text{ref}}} \frac{\partial}{\partial z} \mathcal{J}_v \qquad -fu \tag{3b}$$

$$\frac{\partial u}{\partial t} = -\frac{1}{\rho_{\text{ref}}} \frac{\partial}{\partial z} \mathcal{J}_{u} + fv \tag{3a}$$

$$\frac{\partial v}{\partial t} = -\frac{1}{\rho_{\text{ref}}} \frac{\partial}{\partial z} \mathcal{J}_{v} - fu \tag{3b}$$

$$\frac{\partial \theta}{\partial t} = -\frac{1}{\rho_{\text{ref}}} \frac{\partial}{\partial z} \mathcal{J}_{\theta} + \frac{1}{\rho_{\text{ref}} c_{p}} \frac{\partial}{\partial z} Q_{\text{sr}} \tag{3c}$$

$$\frac{\partial S}{\partial t} = -\frac{1}{\rho_{\text{ref}}} \frac{\partial}{\partial z} \mathcal{J}_S \tag{3d}$$

The four resolved liquid ocean prognostic variables in NEMO are the horizontal velocity components u and v, the potential temperature  $\theta$ , and the practical salinity S. As for the atmosphere, we use  $\partial_z \mathcal{J}_{\phi}$  to denote the vertical turbulent transport. The vertical turbulent fluxes are expressed using eddy viscosities and diffusivities, which are a function of a prognostic turbulent kinetic energy (TKE) and a diagnostic mixing length. The Boussinesq reference density is  $\rho_{\rm ref}=1035\,{\rm kg\,m^{-3}}$  and a non-linear equation of state is used to compute the Brunt-Väisälä frequency entering the computation of the TKE equation.  $Q_{\rm sr}(z)$  is the penetrative part of the solar radiative flux. As before, f is the Coriolis parameter and  $c_p$  is a constant specific heat capacity of seawater. For the numerical simulations presented in Section 4, we use two versions of the EC-Earth AOSCM: EC-Earth 3 and a development version of EC-Earth 4. In version 3 of the AOSCM, a *linear free surface* is employed, implying that the volume of the ocean column remains constant. In contrast, the AOSCM version of EC-Earth 4 uses a nonlinear free surface, allowing the volume to vary in response to freshwater fluxes<sup>1</sup>. This change affects the boundary condition for the salinity equation (3d) at the sea surface, as we will discuss later.

NEMO integrates a sea ice component called the Louvain-La-Neuve sea Ice Model (LIM3) for versions prior to NEMO 4 and Sea Ice modelling Integrated Initiative (SI<sup>3</sup>) for later versions. The differences between LIM3 and SI<sup>3</sup> are primarily at the level of the software environment. In the SCM formalism, only ice thermodynamics (i.e., all the processes controlling the increase or decrease of sea ice mass) are involved, the dynamics are assumed to be at rest. The model is based on the assumption that sea ice is a two-phase, two-component porous medium (mushy layer) covered by one or multiple layers of snow. Since thermodynamical properties strongly depend on thickness, the ice pack is modeled in terms of how its mass is distributed across a number of thickness categories, and how this distribution evolves in time and space (multi-category framework). Vertical heat conduction and storage in sea ice are described by a heat equation, which is adapted to account for the internal absorption of solar radiation. The boundary conditions for the heat equation follow from an energy balance at the surface and at the ice base to compute the sea ice surface temperature  $T^i$  and vertical conduction fluxes, respectively. At the ice base the temperature is prescribed as a Dirichlet condition and assumed to be at the local freezing point. Since the surface fluxes depend nonlinearly on  $T^i$ , the surface energy balance is typically solved using a Newton-Raphson method. This requires an estimate  $\partial Q_{\rm ns}/\partial T^i$ , which is provided by the atmospheric model (cf. Figure 1). For more details on the sea ice model formulation, the interested reader can refer to Vancoppenolle et al. (2023).

<sup>&</sup>lt;sup>1</sup>In NEMO terminology, the case of a nonlinear free surface corresponds to the so-called VVL (Vertical Varying Layers) case. In recent versions of the code, the VVL terminology has been replaced by QCO (Quasi-Eulerian COordinate).

Figure 1. Overview of exchanged data between OpenIFS and NEMO during a coupled run of the EC-Earth AOSCM. Technically, OpenIFS sends the *total* fluxes and those over ice, and NEMO internally computes the flux over ocean, e.g.,  $\mathcal{E}_o = \mathcal{E}_{tot} - \mathcal{E}_i$ . NEMO additionally sends sea ice and snow thicknesses, but these are not used by OpenIFS. We have omitted these details here for clarity. The conversion from potential temperature  $\theta$  to absolute temperature T happens inside NEMO before sending the values to OpenIFS. Notation introduced in Section 2.3.

#### 145 2.3 Interface Boundary Conditions



Figure 1 summarizes the variables which are exchanged between OpenIFS and NEMO. These are used to compute the boundary conditions at the air-sea interface, which we present in this section. We do not discuss the boundary conditions at the sea floor and top of atmosphere. In this section and from now on, we use superscripts a, o, and i to distinguish between atmosphere, ocean, and ice quantities whenever necessary.

Boundary conditions are necessary for all vertical turbulent fluxes  $\mathcal{J}_{\phi}$ , the solar surface heat flux  $Q_{\mathrm{sr}}$ , and the net radiative fluxes  $\mathcal{F}_{\mathrm{SW}}$  and  $\mathcal{F}_{\mathrm{LW}}$ . For the radiative fluxes, OpenIFS distinguishes between downward and upward radiation  $\mathcal{F} = \mathcal{F}^{\downarrow} - \mathcal{F}^{\uparrow}$ . A boundary condition at the sea surface is only required for the upward components  $\mathcal{F}_{\mathrm{SW}}^{\uparrow}$ ,  $\mathcal{F}_{\mathrm{LW}}^{\uparrow}$ .

Although the equations for the atmosphere are given in pressure coordinates, we switch to the z-coordinate at the sea surface for consistency between both models. We adopt the following notation: the lowest grid point in the atmospheric grid  $z_1$  is located at  $z \approx 10$ m. We will refer to this point as  $z_1$ . The center of the surface grid cell in NEMO is located at  $z \approx -0.5$ m, which we will refer to as  $z_{-1}$ . Except for  $z_1$  and  $z_{-1}$ , we write down fully continuous boundary conditions here.

## 2.3.1 Interface Boundary Conditions in Absence of Sea Ice

If no sea ice is present and waves are neglected, OpenIFS and NEMO see the same vertical turbulent momentum flux at the sea surface. The resulting boundary condition is given by

$$\mathcal{J}_{M}^{a}|_{z=0} = \mathcal{J}_{M}^{o}|_{z=0} = \rho^{a} C_{M,o} |U(z_{1})|^{2} \frac{u^{a}(z_{1})}{\|u^{a}(z_{1})\|},$$
 (4)

with the vectors  $\mathcal{J}_M = (\mathcal{J}_u, \mathcal{J}_v)^T$ ,  $\mathbf{u} = (u, v)^T$ , and  $C_{M,o}$  the transfer coefficient, a scalar factor which nonlinearly depends on  $\mathbf{u}$ .<sup>2</sup> Furthermore,

$$|U(z_1)| = \sqrt{\|\boldsymbol{u}^a(z_1)\|_2^2 + w_*^2},$$
(5)

is the wind speed, with  $w_*$  the free convection velocity scale which depends on air temperature and moisture near the surface (ECMWF, 2014, §IV.3). Ocean surface currents  $u^o(z_{-1})$  are not taken into account in this boundary condition, which amounts to neglecting the wind-current feedback.<sup>3</sup>

The boundary conditions for  $\mathcal{J}_T$ ,  $\mathcal{J}_q$ ,  $\mathcal{F}_{SW}$ ,  $\mathcal{F}_{LW}$ , and  $Q_{sr}$  are derived by requiring conservation of internal energy at the sea surface. We can write the balance of internal energy at the surface as<sup>4</sup>

$$\mathcal{J}_{T}^{o}|_{z=0} + Q_{\rm sr}|_{z=0} = \mathcal{J}_{T}^{a}|_{z=0} + \mathcal{J}_{q}|_{z=0} + \mathcal{F}_{\rm LW}|_{z=0} + \mathcal{F}_{\rm SW}|_{z=0}. \tag{6}$$

The terms originating from the atmosphere on the right-hand side are mapped to the NEMO fluxes as

$$\mathcal{J}_{T}^{o}|_{z=0} = \underbrace{\mathcal{J}_{T}^{a}|_{z=0} + \mathcal{J}_{q}|_{z=0} + \mathcal{F}_{LW}|_{z=0}}_{Q_{ns}|_{z=0}}$$
(7a)

$$Q_{\rm sr}|_{z=0} = \mathcal{F}_{\rm SW}|_{z=0},$$
 (7b)

where we have introduced the nonsolar heat flux  $Q_{\rm ns}$ . In contrast to  $Q_{\rm sr}$ , it does not penetrate below the ocean surface. NEMO receives these fluxes from OpenIFS, where they are computed as

$$\mathcal{J}_{T}^{a}|_{z=0} = \rho^{a} c_{p}^{a} C_{H,o} |U(z_{1})| (T^{a}(z_{1}) - SST)$$
 (8a)

$$\mathcal{J}_{q}^{a}\big|_{z=0} = \rho^{a} c_{p}^{a} C_{H,o} |U(z_{1})| \left(q^{a}(z_{1}) - 0.98 q_{\text{sat}}(\text{SST})\right) \tag{8b}$$

$$\mathcal{F}_{\text{LW}}^{\uparrow}\Big|_{z=0} = \varepsilon^o \sigma \text{SST}^4$$
 (8c)

$$\left. \mathcal{F}_{\mathrm{SW}}^{\uparrow} \right|_{z=0} = \alpha^{o} \left. \mathcal{F}_{\mathrm{SW}}^{\downarrow} \right|_{z=0}$$
 (8d)

Therein,  $C_{H,o}$  is the transfer coefficient for sensible heat. The sea surface temperature (SST) is computed from  $T^o(z_{-1})$  inside OpenIFS by taking into account the cool skin effect (ECMWF, 2014, §IV.8.9). The saturation humidity at the surface  $q_{\rm sat}$  is defined in terms of the SST. In the equations for the upward radiative fluxes,  $\sigma$  denotes the Stefan-Boltzmann constant, while  $\varepsilon^o$  and  $\alpha^o$  denote the ocean emissivity and albedo, respectively. Values for these are specified in OpenIFS, they are not computed by NEMO.

When a linear free surface is considered (i.e. in the AOSCM version of EC-Earth 3), the boundary condition for  $\mathcal{J}_S$  is computed as a virtual salt flux (Huang, 1993)

$$\mathcal{J}_S|_{z=0} = (\mathcal{E} - \mathcal{P})S(z_{-1}),\tag{9}$$

<sup>&</sup>lt;sup>2</sup>This way of approximating  $\mathcal{J}_{\phi}$  is based on boundary layer theory for stratified fluids (Olbers et al., 2012; Monin and Obukhov, 1954).

<sup>&</sup>lt;sup>3</sup>This has implications on the stability and accuracy of the coupling scheme, cf. Connors and Ganis (2011); Renault et al. (2019).

<sup>&</sup>lt;sup>4</sup>This formulation does not contain three additional terms for internal energy exchange due to precipitation and evaporation. As stated in Olbers et al. (2012, p. 54), these are comparatively small and "can usually be ignored."

wherein  $\mathcal{P}$  and  $\mathcal{E}$  are the total precipitation and evaporation fluxes computed by OpenIFS. In the non-linear free surface case (i.e. in the AOSCM version of EC-Earth 4), the  $(\mathcal{E}-\mathcal{P})$  flux is accounted for as mass transport in the free-surface evolution equation  $(\partial_t \eta = -(\mathcal{E}-\mathcal{P})/\rho_0)$  and  $\mathcal{J}_S|_{z=\eta} = 0$ , with  $\eta$  the sea surface height.

## 190 2.3.2 Full Sea Ice Cover



We now give the boundary conditions for the atmosphere and ocean in case the ocean surface is fully covered by sea ice. This case does not actually appear in practice, since the sea ice area fraction in NEMO has a maximum value smaller than 100%, specified at runtime. However, it is useful to consider this case before defining the boundary conditions for a partially ice-covered ocean surface.

We begin with kinetic energy: The turbulent momentum flux boundary conditions are given by

$$\mathcal{J}_{M}^{a}\big|_{z=0} = \rho^{a} C_{M,i} |U(z_{1})|^{2} \frac{\boldsymbol{u}^{a}(z_{1})}{\|\boldsymbol{u}^{a}(z_{1})\|}$$
(10a)

$$\left. \mathcal{J}_{M}^{o} \right|_{z=0} = \rho^{o} C_{D,i} \left\| \boldsymbol{u}^{i} - \boldsymbol{u}^{o}(z_{-1}) \right\| \left( \boldsymbol{u}^{i} - \boldsymbol{u}^{o}(z_{-1}) \right).$$
 (10b)

The turbulent momentum flux seen by the atmosphere  $\mathcal{J}_{M}^{a}$  is almost the same as in the ice-free case, differing only in the transfer coefficient, since different surface roughness lengths are assumed for the ice and ocean surfaces. On the other hand, NEMO now computes its own turbulent momentum flux beneath ice  $\mathcal{J}_{M}^{o}$ , using a separate ocean-ice drag coefficient  $C_{D,i}$  and taking into account horizontal ice velocity. In the SCM,  $u^{i} \equiv 0$ , which simplifies the boundary condition in practice.

For the sensible and latent heat fluxes seen by the atmosphere, as well as the upward components of radiative fluxes, one obtains

$$\mathcal{J}_{T}^{a}|_{z=0} = \rho^{a} c_{p}^{a} C_{H,i} |U(z_{1})| \left(T^{a}(z_{1}) - T^{i}\right)$$
(11a)

$$\mathcal{J}_q|_{z=0} = \rho^a c_p^a C_{H,i} |U(z_1)| \left( q^a(z_1) - q_{\text{sat}}(T^i) \right)$$
(11b)

$$\mathcal{F}_{\text{LW}}^{\uparrow}\Big|_{z=0} = \varepsilon^i \sigma T^{i^4} \tag{11c}$$

$$\mathcal{F}_{\mathrm{SW}}^{\uparrow}\Big|_{z=0} = \alpha^i \, \mathcal{F}_{\mathrm{SW}}^{\downarrow}\Big|_{z=0}$$
 (11d)

The structure is equivalent to the corresponding boundary conditions in absence of sea ice. However, OpenIFS now uses the ice surface temperature  $T^i$  instead of SST, as well as a different transfer coefficient, surface emissivity, and albedo. Both  $T^i$  and  $\alpha^i$  are coupling variables provided by the sea ice model as the weighted mean over sea ice categories.

The remaining ocean boundary conditions beneath sea ice have the form

$$Q_{\rm sr}|_{z=0} = \mathcal{F}_{\rm SW}|_{z=0} e^{-\kappa h} \tag{12a}$$

$$\mathcal{J}_T^o|_{z=0} = Q^i \tag{12b}$$

$$\left. \mathcal{J}_S \right|_{z=0} = \mathcal{S}_t. \tag{12c}$$

The solar radiation boundary condition takes into account that solar radiation attenuates exponentially in sea ice, following the Beer-Lambert law with extinction coefficient  $\kappa$  and sea ice thickness h. The ice-ocean heat flux  $Q^i$ , as well as the salt flux due to sea ice growth and melt  $\mathcal{S}_t$ , are computed by the sea ice model.

#### 2.3.3 Nonzero Sea Ice Cover

In case the ocean is partly covered by sea ice, the boundary conditions from Sections 2.3.1 and 2.3.2 are linearly combined based on the sea ice area fraction  $a_i \in [0,1]$ . For instance for the turbulent momentum fluxes seen by the atmosphere,

$$\left. \mathcal{J}_{M}^{a} \right|_{z=0} = a_{i} \left. \mathcal{J}_{M,i}^{a} \right|_{z=0} + (1 - a_{i}) \left. \mathcal{J}_{M,o}^{a} \right|_{z=0}.$$
 (13)

Here we use subscripts o, i to denote the ice-free and ice-covered boundary conditions from Section 2.3.1 and Section 2.3.2, respectively. The same approach is used for  $\mathcal{J}_{M}^{o}$ ,  $\mathcal{J}_{T}^{a}$ ,  $\mathcal{J}_{q}$ ,  $\mathcal{F}_{LW}^{\uparrow}$ ,  $\mathcal{F}_{SW}^{\uparrow}$ , and  $Q_{sr}$ . The two remaining boundary conditions for the ocean are given by

$$\mathcal{J}_T^o|_{z=0} = (1-a_i)Q_{\text{ns},o} + Q^i$$
 (14a)

$$\mathcal{J}_S|_{z=0} = (\mathcal{E}_o - \mathcal{P})S(z_{-1}) + \mathcal{S}_t. \tag{14b}$$

 $SI^3$  takes into account the ice area fraction in computing  $Q^i$  and  $\mathcal{S}_t$ . Again, the boundary condition (14b) is the one used in the linear free surface case, while in the non-linear case the contribution from  $(\mathcal{E}_o - \mathcal{P})$  vanishes from  $\mathcal{J}_S|_{z=0}$  and instead appears in the free-surface evolution equation.

#### 230 3 Coupling Algorithms


To compute the interface boundary conditions presented in the previous section, OpenIFS and NEMO exchange the coupling variables from Figure 1 at discrete points in time. The coupling algorithm implemented in an ESM specifies the numerical method used for this data exchange, e.g., the order of communication or whether data is averaged in time. In this section, we present standard coupling algorithms in ESMs and introduce SWR as an iterative coupling approach. Finally, we explain our implementation approach for switching between coupling algorithms in the EC-Earth AOSCM.

#### 3.1 Standard Coupling Algorithms

NEMO and OpenIFS exchange the coupling data from Figure 1 at N+1 coupling time steps  $t_n$ , where  $n=0,\ldots,N,$   $t_0=0,$   $t_N=\mathcal{T}$  the total simulation time. We call the intervals  $[t_n,t_{n+1}]$  coupling windows and assume a constant coupling time step size  $\Delta t_{\rm cpl}=t_{n+1}-t_n=\mathcal{T}/N$ . The model components can use time step sizes smaller than  $\Delta t_{\rm cpl}$ . Commonly, the atmosphere component uses smaller time step sizes than the ocean component.

The default coupling algorithm in EC-Earth and the EC-Earth AOSCM is the parallel coupling algorithm, depicted in Figure 2a. At a coupling time step  $t_n$ , the coupler sends the time average of a given coupling variable over all model time steps of the previous coupling window  $[t_{n-1}, t_n]$ . This value is used for the interface boundary conditions in all model time

(a) The parallel coupling algorithm.

(b) The sequential atmosphere-first coupling algorithm.

Figure 2. The parallel and sequential atmosphere-first coupling algorithms. Coupling variables are averaged in time (denoted by  $\langle \cdot \rangle$ ) and sent at coupling time steps  $t_n$ . Coupling data received at  $t_n$  is used as a boundary condition in the following coupling window  $[t_n, t_{n+1}]$ .

steps of the next coupling window  $[t_n, t_{n+1}]$ , introducing a coupling lag: The boundary conditions used to advance OpenIFS and NEMO from  $t_n$  to  $t_{n+1}$  are computed using quantities averaged between  $t_{n-1}$  and  $t_n$ . All standard coupling schemes in ESMs are lagged, albeit to a different extent. In the multiphysics coupling literature, this setup is referred to as a *multirate* discretization with a *loose* coupling algorithm (Keyes et al., 2013).

It is also possible to use a sequential coupling algorithm, as done, e.g., by the ECMWF (Mogensen et al., 2012). As the name suggests, the submodels are run after each other in this configuration. Two variants exist, the sequential atmosphere-first and the sequential ocean-first algorithms, the former depicted in Figure 2b. In practice, the sequential ocean-first algorithm is not used by state-of-the-art general circulation models (Marti et al., 2021, ).

In Marti et al. (2021), sequential coupling schemes led to improved numerical results for a 3D coupled general circulation model compared to the parallel algorithm, with the sequential atmosphere-first version outperforming the ocean-first algorithm. However, note that if the computational effort for both models is similar, switching from parallel to sequential coupling increases the time to solution roughly by a factor of two. For a comparison of parallel and sequential coupling approaches, we refer to, e.g., Mehl et al. (2016); Meisrimel and Birken (2022).

#### 3.2 Schwarz Waveform Relaxation






Atmosphere-ocean coupling can be seen as an example of domain decomposition with no overlap of the subdomains (Lemarié et al., 2014; Gross et al., 2018). With this perspective, the parallel and sequential coupling algorithms correspond to the first iteration of a Schwarz waveform relaxation (SWR) method. This is an iterative coupling approach where the submodels successively update each other's boundary data, as depicted in Figure 3a. In iteration k and coupling window  $[t_n, t_{n+1}]$ , a component reads coupling data from the same coupling window but the previous iteration, k-1. In case SWR converges, one obtains a solution without the coupling lag in the limit  $k \to \infty$ , cf. Figure 3b.

SWR converges under certain conditions on the well-posedness of the underlying coupled problem, in particular Lipschitz-continuity of right-hand sides (Janssen and Vandewalle, 1996), here given in Equations (1) and (3), and a correct choice of interface boundary conditions (Gander, 2006). In case of convergence, one can use it as a tool to quantify the coupling error

Figure 3. SWR for two subdomains depicted by the data dependency across iterations (a) and in the converged limit (b).

of standard coupling algorithms, as illustrated in the previous study by Marti et al. (2021): by repeating the time integration until the boundary data converges, a reference solution is obtained. This can be compared to the results with standard coupling schemes, where the coupling lag introduced numerical errors.

In case of non-convergence, the underlying coupled problem might not "obey regularity" or could even be ill-posed (Gross et al., 2018, p. 3523). Such a result would not be surprising, given that ESM components are typically developed independently. In this case, or if SWR converges slowly, domain decomposition theory implies that the formulation of the coupled problem should be revisited (Lemarié et al., 2014). Particularly, subgrid-scale physics parameterizations are sometimes based on discontinuous (semi-empirical) formulations which impair the differentiability of right-hand sides and the regularity of solutions, cf. Gross et al. (2018, §5).

## 3.3 Implementation in the EC-Earth AOSCM




OpenIFS and NEMO are coupled with the OASIS3-MCT coupling software (Craig et al., 2017), which we refer to as OASIS. It exchanges data between model components and can, in that process, apply various transformations to the coupling fields, e.g., time-averaging or regridding. Model components call OASIS to instantiate and finalize the coupler, declare coupling fields, and trigger communication.

During a coupled simulation, OASIS determines whether data has to be sent at a given model time step, based on user input provided in the configuration file *namcouple*. Many aspects of the coupling setup can be changed using this file, including the coupling window size and transformation options. Overall, essential parts of the coupling logic are invisible to the model components.

Whether an AOSCM experiment uses the parallel algorithm or one of the sequential ones at runtime can be controlled by modifying the LAG parameters in *namcouple*. This was previously mentioned and used in Marti et al. (2021); Streffing et al. (2022). In Section A, we explain how to set these parameters based on the desired coupling scheme.

To implement SWR in the EC-Earth AOSCM, we follow an approach first used for the CNRM-CM6-1 coupled SCM (Valcke, 2021; Voldoire et al., 2022). Here, OASIS is utilized to support repeated evaluations of the same time interval. Runtime settings of OASIS are adapted using external scripts which take control of running coupling iterations. This is in

contrast to the implementation of Marti et al. (2021), where the main time loops of the atmosphere and ocean models were adjusted significantly to support "rewinding" in time. Designing the implementation based on the coupling software, with minimal changes in the ocean and atmosphere models, allows to naturally extend the basic SWR algorithm, e.g., by black-box acceleration methods such as a relaxation step or Quasi-Newton approaches (Rüth et al., 2021). More importantly, it is easier to reuse in other models, since OASIS is widely used in the climate community (Craig et al., 2017).

The pseudocode for this approach is given in Algorithm 1. The first iteration is equivalent to a regular coupled AOSCM run, with the same available settings (including switching between the parallel or one of the sequential algorithms). During the simulation, OASIS saves the values of the interface variables at every coupling time step to an output file. In consecutive iterations, the models instead read in coupling variables from the output file produced in the previous iteration (after some postprocessing). This is achieved by using a different namcouple file as soon as the iteration number k is larger than one.

Our SWR implementation thus runs the model executable *multiple times* with different configuration settings. Each call to the compiled model corresponds to one iteration. An outer layer written in Python controls the SWR algorithm, making it a runtime feature which is kept separate from the model code. The implemented solution is equivalent to SWR with piecewise constant interface data averaged over each coupling window  $\Delta t_{\rm cpl}$ , with the Schwarz window size equal to the simulation time  $\mathcal{T}$ . Since the number of SWR iterations necessary to converge generally grows with  $\mathcal{T}$ , this approach is likely unsuitable for very long simulations. However, we will see in the next section that much can be learned about coupling error sources in a given model, even when considering short time scales.

## Algorithm 1 SWR Algorithm implemented in the EC-Earth AOSCM

```
\begin{aligned} k &\leftarrow 1 \\ \mathbf{while} \ k &< K \ \mathbf{do} \\ \mathbf{choose\_namcouple}(k) \\ t &\leftarrow 0 \\ \mathbf{while} \ t &< T \ \mathbf{do} \\ \mathbf{for} \ \mathbf{model} \ \mathbf{in} \ [\mathbf{OpenIFS}, \ \mathbf{NEMO}] \ \mathbf{do} \\ \mathbf{model.read\_coupling\_data}(t) \\ \mathbf{model.integrate}(t, \Delta t) \\ \mathbf{model.write\_coupling\_data}(t) \\ \mathbf{end} \ \mathbf{for} \\ t &\leftarrow t + \Delta t \\ \mathbf{end} \ \mathbf{while} \\ \mathbf{postprocess\_iteration}(k) \\ \mathbf{end} \ \mathbf{while} \end{aligned}
```



In the AOSCM, we terminate the SWR algorithm either after a fixed number of iterations or using a runtime termination criterion. For the latter, consider  $c^k \in \mathbb{R}^N$  to be the vector of values of a coupling variable c in iteration k. It has dimension N,

equal to the number of coupling windows in the simulation. As a runtime termination criterion we use

$$\left\|c^{k} - c^{k-1}\right\|_{\infty} \le \text{TOL}\left\|c^{1}\right\|_{\infty},\tag{15}$$

where TOL > 0 is the relative tolerance. The termination criterion is satisfied once the inequality holds for all coupling variables given in Figure 1.<sup>5</sup>

In cases where we use a fixed number of iterations K, with K large enough to achieve convergence up to numerical precision, we will display the relative error with respect to the final iterate

$$e_{\text{rel}}^{k} = \frac{\left\| c^{k} - c^{K} \right\|_{\infty}}{\left\| c^{1} \right\|_{\infty}}.$$
 (16)

## 4 Numerical Results

In this section, we present a range of numerical results investigating the SWR convergence behavior and coupling error of the EC-Earth AOSCM. We pick three model versions for our experiments: OpenIFS cy40r1–NEMO 3.6; OpenIFS cy43r3–NEMO 3.6; and OpenIFS cy43r3–NEMO 4.0.1. The former corresponds to the EC-Earth 3 AOSCM as published in Hartung et al. (2018), while the latter two are development versions on the path to the EC-Earth 4 AOSCM. All experiments in this section were carried out with all model versions. If behavior does not change qualitatively between versions, we only report results from the base version (OpenIFS cy40r1–NEMO 3.6).

For all experiments in this paper, we used time step sizes as given in Table 1. These are comparable to previous experiments with the AOSCM (Hartung et al., 2018). In particular for the atmosphere, the radiation scheme is called in every time step, while the turbulence parameterization is called twice per time step. We have chosen a larger coupling time step to reflect standard EC-Earth or ECMWF setups.

For the vertical resolution, we use two different grids in the atmosphere, predetermined by the available input data. In both cases, grid resolution is about  $20\,\mathrm{m}$  near the air–sea interface and decreases towards the top of the atmosphere (the last grid cell at  $80\,\mathrm{km}$  spans multiple kilometers). The vertical grid in the ocean also varies with depth: grid thickness is about  $1\,\mathrm{m}$  near the surface and increases with depth to reach  $200\,\mathrm{m}$  at the bottom. The number of model levels is given in Table 1.

#### 4.1 Ice-Free Sea Surface



As an example for atmosphere-ocean coupling, we consider the PAPA station in the north-eastern Pacific (nominally at 50°N, 145°W) in July 2014. This case was part of the original EC-Earth AOSCM paper, since there is a long history of studying physics parameterizations in the ocean at this buoy (Hartung et al., 2018). We consider two experiments: (a) a single four-day

<sup>&</sup>lt;sup>5</sup>Marti et al. (2021) determine SWR convergence based on the SST, since this is the only coupling variable the ice-free ocean sends to the atmosphere. This criterion is clearly not sufficient in case of sea ice. There is another argument to consider for the ice-free case: the atmosphere "by construction computes the same fluxes" (Marti et al., 2021, p. 2972) once the SST converges. Numerically, this holds when the difference across iterations is at values near machine precision. However, they (and we) use convergence tolerances based on whether differences are physically negligible. It is unclear a priori if a physically negligible change in SST might cause (or be caused by) a non-negligible change of atmospheric fluxes.

|                         | PAPA             | MOSAiC           |
|-------------------------|------------------|------------------|
| $\Delta t_{ m cpl}$     | $60\mathrm{min}$ | $60\mathrm{min}$ |
| $\Delta t_{ m o}$       | $15\mathrm{min}$ | $15\mathrm{min}$ |
| $\Delta t_{ m i}$       | _                | $15\mathrm{min}$ |
| $\Delta t_{ m a}$       | $15\mathrm{min}$ | $15\mathrm{min}$ |
| Atmosphere model levels | 60               | 137              |
| Ocean model levels      | 75               | 75               |

**Table 1.** Grid setup for the numerical experiments.  $\Delta t_{\rm atm}$  includes the radiation time step in our setup.

experiment starting 1 July 2014, 00:00 UTC and (b) a set of 112 two-day experiments starting between 1 July 2014, 00:00 UTC and 28 July 2014, 18:00 UTC, spaced 6 h apart.

As initial and forcing data for OpenIFS, we use a forcing file based on 6-hourly ERA-Interim data for the duration 1 – 30 July 2014. This forcing file is distributed with the EC-Earth AOSCM. NEMO is initialized using daily values we extracted from the CMEMS Global Ocean Physics Reanalysis (European Union-Copernicus Marine Service, 2018) at the location of the station, interpolated to the 75 vertical levels used by the model.

#### 4.1.1 SWR Convergence Results




In case a coupled problem is well-posed, we expect convergence of the SWR algorithm. For the four-day experiment, we test this by measuring the relative error with respect to the final iterate as given in Equation (16). Figure 4 shows the SWR convergence behavior for four coupling variables, representative for the data exchanged between OpenIFS and NEMO in absence of sea ice: the SST (sent by NEMO), as well as  $Q_{\rm ns}$ ,  $\tau_x$ , and  $\mathcal{P}$  (sent by OpenIFS).

The relative error stays near-constant for about seven iterations, before decaying roughly linearly every other iteration. Such behavior is expected for parallel SWR with long Schwarz windows (e.g., Gander, 2006; Janssen and Vandewalle, 1996). The figure shows that the error for SST stays about two orders of magnitude below the relative error for atmospheric data. Once the relative error of the SST reaches its numerical convergence limit, the atmospheric variables also taper off. This happens around iteration 22.

Such a large number of iterations is infeasible for long or many runs, especially in 3D. One way to reduce iterations is to terminate the SWR algorithm once the differences between iterations become physically negligible. For this we use the termination criterion (15) with a relative tolerance of  $TOL = 1 \times 10^{-5}$ . In the 4-day experiment, this happens after 13 iterations, cf. Figure 4.

We used the same criterion for the 112 two-day-experiments distributed throughout July 2014. Out of all simulations, 109 satisfied the termination criterion in less than 30 iterations. The number of iterations necessary to satisfy (15) is distributed according to Figure 5, with a mean and median iteration count of 11.4 and 12, respectively. These results indicate how fast the SWR algorithm converges in practice in the absence of sea ice.

Figure 4. Relative Error  $e_{\rm rel}$  for each SWR iteration in the four-day experiment.. The dashed line marks the tolerance used for the termination criterion in later experiments.

Figure 5. SWR iteration count until termination criterion (15) is satisfied.

For the experiments which did not satisfy the termination criterion after 30 iterations, increasing the maximum number of iterations did not help to satisfy the runtime termination criterion. In these cases, the output entered a (small) oscillation, similar to observations in Marti et al. (2021). The fraction of non-converged experiments was small (below 10%) but varied between the versions of the AOSCM.

## 4.1.2 Coupling Error Estimation with SWR

We now estimate the coupling error of the parallel, atmosphere-first, and ocean-first algorithms. We return to the four-day-experiment at the PAPA station and consider iteration 30 as the reference solution. Figure 6 presents simulation results for all four coupling algorithms. The first three panels show thermodynamic prognostic variables at the grid point closest to the air-sea interface: atmospheric temperature  $T^a$  and humidity q at  $z_1 = 10 \,\mathrm{m}$ , and the SST as computed by NEMO, i.e.,  $T^o(z_{-1})$ .

Figure 6. Thermodynamic prognostic variables, as well as surface heat fluxes, in the 4d-experiment for different coupling algorithms.

For these variables, model physics have significant impact and proper two-way coupling is implemented in the AOSCM (as opposed to wind speeds, ocean currents, and salinity).

The output for  $T^a$  and q shows significant differences between the coupling schemes. For instance, the temperature coupling error reaches about 1 °C in magnitude for all algorithms at some point during the simulation. This starts on 2 July at daytime, where the two sequential algorithms perform worse than the parallel one. Around midnight, the parallel result separates from the SWR solution and joins the other two. At the end, the coupling error of the parallel algorithm is largest. This is similar for q, although the parallel algorithm stays close to the SWR result for longer, until 3 July around noon.

For the SST, the parallel and ocean-first coupling schemes result in a phase shift of 1 h compared to the atmosphere-first and reference solution. Such a phase shift occurs for all prognostic ocean variables (not shown) and has been observed and explained in Marti et al. (2021). Additionally, all non-iterative coupling schemes give slightly warmer results, most prominent for the parallel and ocean-first algorithms. The coupling error is largest on 3 July 2014 and decreases again afterwards.

Overall, the SST coupling error is small compared to  $T^a$  and q, related to atmospheric heat fluxes. We illustrate this in Figure 6d-e with the sum of turbulent and radiative heat fluxes at the surface,  $\mathcal{J}_T^a|_{z=0} + \mathcal{J}_q|_{z=0}$  and  $\mathcal{F}_{LW}|_{z=0} + \mathcal{F}_{SW}|_{z=0}$ ,

**Figure 7.** Vertical temperature profile in the atmosphere for the four-day-experiment at the PAPA station.




respectively. On 2 July, the sequential algorithms produce significantly lower heat fluxes than the parallel and SWR solution, thus cooling the column. Notably, the sequential algorithms produce an almost fully cloud-covered column at this point, while the other two coupling schemes give a total cloud cover below 50% (not shown). The parallel solution departs from the reference solution around midnight on 3 July, clearly visible for the turbulent heat flux. The heat flux differences are very small in the second half of the simulation, which is why the coupling error in  $T^a$  and q does not decrease after 3 July 2014.

In ice-free conditions, switching the coupling algorithm corresponds to a small (cf. Figure 6c) perturbation in the SST seen by the atmosphere. The results above emphasize that the atmospheric thermodynamics react strongly to this, but not why. We suppose that this is due to parameterizations in OpenIFS which yield a discontinuous right-hand side of the primitive equations.

An example is the convective mass flux parameterization, which is part of the vertical turbulent fluxes (2) in the atmosphere. In OpenIFS, a switch-case statement is evaluated in every time step to determine the type and height of the boundary layer based on surface properties, affecting vertical temperature and humidity profiles (Fitch, 2022). Indeed, the vertical temperature profiles differ strongly after 2 July, while turning off the mass flux scheme (namelist parameter LECUMF) yields a very small coupling error, cf. Figure 7. Ultimately, this four-day experiment illustrates a strong sensitivity of atmospheric convection to small variations in the surface boundary conditions, leading to a large coupling error.

To get a more representative estimate of the coupling error, we now consider the 110 two-day simulations where the SWR algorithm successfully terminated in less than 30 iterations. The first iteration satisfying the termination criterion is chosen as the reference solution. We compute the coupling error at the end of the simulation for SST,  $T^a$ , and q, taking the 2-norm over the atmospheric boundary layer for the latter two:

$$e_{j}(SST) = |SST_{j} - SST_{SWR}|$$

$$400 e_{j}(T^{a}) = ||T_{j}^{a} - T_{SWR}^{a}||_{2}$$

$$e_{j}(q) = ||q_{j} - q_{SWR}||_{2}.$$
(17)

Therein,  $j \in \{a, o, p\}$  denotes the non-iterative coupling scheme (i.e., atmosphere-first, ocean-first, parallel). We do not recompute the size of the boundary layer for each experiment but instead select the lowest ten model levels of OpenIFS, which span from surface pressure down to  $p = (913 \pm 5) \, \text{hPa} \, (z \approx 900 \, \text{m})$ .

The maximum errors across all  $3 \times 110 = 327$  non-iterative simulations are large considering the length of the experiments:

$$e_{\text{max}}(\text{SST}) = 0.58^{\circ}\text{C}$$
$e_{\text{max}}(T^a) = 3.99^{\circ}\text{C}$  (18)
$$e_{\text{max}}(q) = 1.80 \,\text{g/kg}.$$

However, such large coupling errors come up rarely, while the majority of experiments is close to the SWR result. To illustrate this, we compute a weighted error  $e_j/e_{\rm max}$ , group the result into bins, and count how often a coupling scheme appears in each error range.

The results are given in Figure 8, which is intentionally similar to Marti et al. (2021, Fig. 5). The coupling error stays below 10% of the respective  $e_{\text{max}}$  in most of the experiments. We can conclude that the sequential atmosphere-first algorithm clearly outperforms the other two coupling schemes for SST, but not for atmospheric quantities. Note also that large coupling errors  $(e.g., \geq 0.2e_{\text{max}})$  are observed for a considerable number of experiments in the atmosphere, while this is not the case for the ocean. In the study by Marti et al. (2021), it was sufficient to only consider SST output for studying atmosphere-ocean coupling errors. This seems to be a model-dependent result and is not the case for the EC-Earth AOSCM, where atmosphere and ocean variables show fundamentally different coupling scheme performance.<sup>7</sup>

#### 4.2 Ice-Covered Sea Surface

To study atmosphere-ocean-sea ice coupling in the AOSCM, we consider the case of the YOPP targeted observation period (TOP) in the Arctic during the MOSAiC expedition in April 2020 (Shupe et al., 2022; Svensson et al., 2023). During this period, two warm air intrusions were observed, transient events which transport a large amount of heat and moisture. They

<sup>&</sup>lt;sup>6</sup>Note that they used the maximum difference in SST between two subsequent coupling windows, but this is hard to mimic in our implementation where the Schwarz window spans the whole simulation.

<sup>&</sup>lt;sup>7</sup>Remark: The same set of experiments with LECUMF=.false. gave signficantly smaller observed maximum coupling errors for SST and  $T^a$ . Also, in that case the sequential algorithms seem to perform slightly better for atmospheric variables than the parallel algorithm.

Figure 8. Weighted error  $e_i/e_{max}$  for the two-day TOP experiments where SWR terminated, grouped by coupling scheme.

directly affect atmospheric physics and sea ice thermodynamics, namely via the surface energy budget, cloud formation, and vertical turbulence. For this reason, warm air intrusions are suitable to be studied with a coupled SCM, informing both physical understanding and model development.

We specifically study the nine-day period 12 – 20 April 2020 at 84.45°N, 16.0°E. The initial and forcing file for the atmosphere contains hourly data, extracted from the fifth generation ECMWF atmospheric reanalysis (ERA5, Hersbach et al., 2023). Initial data for the ocean and sea ice is extracted from the CMEMS Global Ocean Physics Reanalysis for NEMO and from the Arctic Ocean Physics Reanalysis for LIM and SI<sup>3</sup> (European Union-Copernicus Marine Service, 2018, 2020). In this time period, the ocean is almost completely covered by sea ice, with a mean sea ice concentration of 99.2%. As before, the ocean grid has 75 vertical levels. For LIM and SI<sup>3</sup>, we use five ice categories, two ice layers, and one snow layer.

We carry out two-day experiments starting every two hours between 12 April 2020, 00:00 UTC, and 18 April 2020, 22:00 UTC, giving 84 time periods. The SWR algorithm did not terminate for any of these experiments in the EC-Earth 3 AOSCM with OpenIFS cy40r1 and NEMO 3.6. Instead, oscillations over a large and unrealistic value range develop in this time period. The spread of the first 30 iterations in two exemplary simulations is shown for atmospheric temperature on the lowest model level in the leftmost panels of Figure 9. The first iteration is highlighted and corresponds to the solution produced with the parallel algorithm.



This behavior does not meaningfully change with an upgraded atmosphere component (OpenIFS cy43r3–NEMO 3.6, panels b and e), but it is strongly improved when the NEMO SCM is at version 4.0.1. This indicates that issues with coupling to the sea ice component LIM3 were responsible for this behavior, and that these are largely resolved in NEMO 4 (with sea ice component SI<sup>3</sup>). For the case of Figure 9c, the SWR algorithm even satisfies the termination criterion (15) after 8 iterations, with convergence behavior comparable to Figure 4 (not shown).

**Figure 9.** 30 SWR iterations for 10 m atmospheric temperature and different EC-Earth AOSCM versions, before (a-c) and after (d-f) the first warm air intrusion. First iteration (corresponding to the parallel algorithm) in dashed magenta.

Figure 10 shows  $10 \,\mathrm{m}$  temperature  $T^a(z_1)$  and ice surface temperature  $T^i$  in the final time step over iterations. We have chosen the experiment from 16-18 April 2020 and focus on the base and latest versions of the model (i.e., panels d and f of Figure 9). The base version of the AOSCM (panels a and b) clearly enters an oscillation after about ten iterations, spanning about  $20\,^{\circ}\mathrm{C}$  and  $33\,^{\circ}\mathrm{C}$  for atmospheric and ice surface temperature, respectively. The second row shows results for the latest version of the AOSCM, where the spread of the iterations is strongly reduced ( $1\,^{\circ}\mathrm{C}$  and  $2.5\,^{\circ}\mathrm{C}$ , respectively). Here one does not see the same oscillatory pattern as for the base version of the model. However, the amplitude of the values does not decrease for an increasing number of iterations, i.e., the SWR algorithm is still not convergent.

The unphysical oscillations for the two older model versions make these results essentially unusable for further analysis of the coupling error. Since EC-Earth development efforts are now oriented toward EC-Earth 4, it makes sense to prioritize studying the results obtained with the latest version of the EC-Earth AOSCM, i.e., OpenIFS cy43r3 coupled to NEMO 4.0.1. Therefore, we now focus on understanding the non-convergence issues identified in Figure 9f and evaluating the coupling errors in the cases where SWR terminates (as in Figure 9c).

Figure 10. Last time step of  $T^a(z_1)$  and  $T^i$  in each iteration for the experiments corresponding to Figure 9, panels d and f.




Regarding non-convergence, Figure 11a shows 10 m temperature results with the parallel algorithm for all 84 experiments. Note that these experiments clearly capture the two warm air intrusions during the observation period, (cf. Svensson et al., 2023, Fig. 3). For only 13 out of the 84 experiments, the SWR algorithm satisfied the termination criterion (15) after less than 30 iterations. Namely, these were the *first* 13 experiments, starting between 12 April and 13 April 2020, 00:00 UTC. They satisfied the criterion after a mean (median) of 6.2 (5) iterations and are marked in red in Figure 11. The dashed line marks the end of the last experiment with a terminated SWR iteration.

To illustrate the convergence issues for the remaining 71 experiments, Figure 11b shows the standard deviation  $\sigma(T^a(z_1))$  for the last ten SWR iterations, for each time step and experiment. One can distinguish two different regimes in this figure, data points where  $\sigma \in [1 \times 10^{-6}, 1 \times 10^{-5}]^{\circ} \text{C}$  and data points where  $\sigma \in [1 \times 10^{-4}, 1]^{\circ} \text{C}$ . For the 13 experiments where the termination criterion was satisfied, the standard deviation stays in the first regime. For all remaining experiments, at least one data point lies in the second regime.

It is clear from the first two panels of Figure 11 that the change in convergence behavior coincides with the steady increase in air temperature starting on 14 April. We now consider the SI<sup>3</sup> output in SWR iteration 30, with Figure 11c showing ice surface temperature  $T^i$  over time. Even though the air temperature hardly exceeds 0 °C during the whole period, the heat flux coming from the atmospheric model leads to sea ice melting during the two warm air intrusions. Since none of the first 13 experiments reaches  $T^i = 0$  °C, the plot suggests that convergence issues might be related to melting conditions in SI<sup>3</sup>.

Further investigation of the sea ice albedo in iteration 30 shows significant jumps during the periods where the ice is at melting conditions, see Figure 11d. Indeed, the sea ice albedo parameterization in  $SI^3$  switches between dry and melting conditions based on whether  $T^i < 0$  °C (Vancoppenolle et al., 2023, §7.1.1). The resulting jump in albedo values is discontinuous.

To test whether the albedo parameterization is responsible for non-convergence of the SWR method, we have changed the  $ice_alb()$  routine in  $SI^3$  to a version that replaces discontinuous jumps with a narrow, smooth transition region, as described in Section C. Both the default and regularized albedo parameterizations give comparable physical results for the TOP experiments, cf. Figure C1. However, the SWR algorithm with the regularized albedo parameterization successfully terminated for 81 out of 84 two-day experiments, with a mean (median) of 11.6 (11) iterations. Thus we can conclude that the discontinuity at  $T^i = 0$  °C in the  $SI^3$  albedo computation is responsible for the consistent non-convergence of the SWR algorithm during the warm air intrusions.<sup>8</sup>

To obtain the coupling error (17) at the final time step as in the ice-free case, we can only consider experiments with successfully terminated SWR iterations. With the default parameterization, this is only possible for the first 13 experiments. We have therefore decided to compute the coupling error using the output generated with the regularized albedo parameterization. As before, we consider the first iteration where the termination criterion (15) is satisfied as the reference solution. To compute  $e_j(T^a)$  and  $e_j(q)$  in the atmospheric boundary layer (average height of  $(433 \pm 249)$  m in these experiments), we take the 2-norm of the lowest 18 atmospheric model levels (up to  $(740 \pm 14)$  m). In addition to the SST, we now also compute the coupling error for the ice surface temperature  $T^i$ .

The maximum observed coupling errors for these variables are





$$e_{\text{max}}(\text{SST}) = 1.9 \times 10^{-2} \,^{\circ}\text{C}$$
 $e_{\text{max}}(T^{i}) = 4.66 \,^{\circ}\text{C}$ 
 $e_{\text{max}}(T^{a}) = 7.48 \,^{\circ}\text{C}$ 
 $e_{\text{max}}(q^{a}) = 0.41 \,\text{g/kg}.$ 
(19)

The maximum errors for the SST and atmospheric humidity are very small compared to the results at the PAPA station. This is not surprising considering that (a) the ocean is almost completely covered by sea ice and thus isolated from the fast-changing atmosphere and (b) the Arctic atmosphere is very cold and dry, giving small values, variation, and errors of q.  $T^a$  and  $T^i$ , on the other hand, have substantial maximum coupling errors. In Figure 12a, we show the atmospheric temperature profile for the experiment where  $e(T^a)$  is maximal (17 April 2020, 16:00 UTC); in this case, the atmosphere-first algorithm produces the worst result. As observed in Figure 7, the magnitude of the error is related to the strong sensitivity of the boundary layer parameterizations to surface variables. Note that, since the ice surface temperature is computed from an energy balance, it directly responds to changes in atmospheric surface fluxes. It is thus unsurprising that coupling errors for  $T^i$  are on a similar order of magnitude as those for  $T^a$ , and much larger than for SST.

Finally, Figure 12b—e show the binned, weighted coupling error (corresponding to Figure 8) for these four variables and the three non-iterative coupling schemes. Panels b and c show that the atmosphere-first algorithm produces the best results for

<sup>&</sup>lt;sup>8</sup>In NEMO 3.6/LIM3, modifying the albedo parameterization does not affect the appearing oscillations, implying that these have a different cause.

**Figure 11.** Results for the 84 overlapping two-day experiments during the YOPP TOP. Output marked in red corresponds to the first 13 experiments, where SWR terminated successfully. The dashed red line marks the last time step of the 13th experiment.

Figure 12. (a): Vertical atmospheric temperature profile for the ice-covered experiment with largest coupling error. (b–e): Weighted error  $e_j/e_{\rm max}$  for the 82/84 two-day experiments with regularized albedo parameterization where SWR terminated, grouped by coupling scheme.

ocean and sea ice variables in the EC-Earth AOSCM. A substantial amount of experiments with the other two algorithms gives large relative coupling errors (e.g., more than 25% of experiments have  $e_j(T^i) \ge 0.2e_{\rm max}$ ). As in the ice-free experiments, performance is more evenly distributed for the atmospheric variables (panels d and e). Thus, no conclusion can be drawn in terms of which coupling algorithm to pick to systematically obtain low coupling errors across all model components.

#### 5 Conclusions



This paper set out to study SWR in atmosphere-ocean and atmosphere-ocean-sea ice coupling. We investigated whether the coupling iteration converges in the EC-Earth AOSCM and compared SWR results with standard ESM coupling algorithms. To this end, we have created a Python wrapper that allows model users to switch between the parallel, atmosphere-first, ocean-first, and SWR algorithms at runtime. The implementation is based on the widely used OASIS3-MCT coupling software, instead of relying on the model components themselves.

Experiments in ice-free conditions showed that the SWR algorithm converges consistently, allowing us to produce reference solutions and quantify the coupling error. In agreement with prior research (Marti et al., 2021), the coupling error for ocean variables is small (usually well below  $0.1\,^{\circ}\mathrm{C}$  for two day simulations) and dominated by a phase error related to solar radiation. This can be mitigated by using the atmosphere-first algorithm. The coupling error for atmospheric variables in the EC-Earth AOSCM was significantly larger in our experiments, and similarly distributed for all three non-iterative schemes. We have found a link between the magnitude of this coupling error and the convective mass flux scheme in OpenIFS, which is a discontinuous parameterization and reacts sensitively to small changes in the SST.

This paper particularly contributes to the sparsely studied area of atmosphere-ocean-sea ice coupling. We have written down the interface boundary conditions in a way that includes the sea ice model contributions. Numerical experiments showed that SWR in the EC-Earth 3 AOSCM produced strong and unphysical oscillations in the presence of sea ice (up to 20 °C for 10 m temperature). These are seemingly related to the old version of the sea ice component (LIM3) and much improved in NEMO 4.0.1, which includes the sea ice component SI<sup>3</sup>.

In the latest development version of the AOSCM, the SWR algorithm still does not converge in cases where sea ice is approaching melting conditions. We were able to explain that these convergence issues are caused by jumps in the ice albedo parameterization once the ice surface temperature reaches 0 °C. The albedo jumps between 0.72 and 0.83 and results in SWR oscillations of up to 1 °C in 10 m temperature.

Discontinuous physics parameterizations in ESMs probably make the underlying coupled problem ill-posed, amplifying small perturbations in initial or boundary data. It is of course reasonable to model distinct physical processes such as melting and drying differently. However, our results suggest that smoothening the transitions between these regimes leads to a more robust coupling setup. Non-convergence of the SWR algorithm helps identify which components cause such issues and can thus guide model development.






We have adjusted the existing albedo computation in SI<sup>3</sup> such that it no longer jumps discontinuously between melting and drying conditions. With this alternative implementation, the SWR algorithm converges consistently and allowed us to quantify the coupling error in case of full sea ice cover. In our experiments, the coupling error after two days was large for atmospheric temperature, but also for ice surface temperature. The latter is mainly driven by atmospheric fluxes entering the surface energy balance in the sea ice model. For ocean and sea ice output variables, the atmosphere-first algorithm once again performs best out of the three non-iterative coupling schemes under consideration. The same cannot be said for atmospheric variables, where performance is evenly distributed. We finally note that the fast reaction time of the sea ice component (particularly regarding ice surface temperature and albedo) makes atmosphere-sea ice coupling an inherently different problem than atmosphere-ocean coupling. Strategies to improve the coupling error of the latter (e.g., using the atmosphere-first algorithm as suggested in Marti et al., 2021) might no longer apply in presence of sea ice.

Code and data availability. The Python wrapper developed to change the coupling scheme of the EC-Earth AOSCM is available at https://github.com/valentinaschueller/ece-scm-coupling under the MIT licence. The exact version used to produce the results in this paper is archived on Zenodo under https://doi.org/10.5281/ZENODO.15088146 (Schüller, 2025a), as are input data and scripts to run the model and produce the plots for all the simulations presented in this paper (Schüller, 2025b). Access to the EC-Earth AOSCM source code is licensed to affiliates of institutions which are members of the EC-Earth consortium. More information on EC-Earth is available at https://www.ec-earth.org (last access: 27 February 2025). The use of the EC-Earth AOSCM also requires a (free) OpenIFS license agreement, which can be obtained from ECMWF. The exact AOSCM versions used in this paper are available for checkout at https://dev.ec-earth.org/projects/ecearth3/repository/show/ecearth3/branches/development/2016/r2740-coupled-SCM/branches/coupling\_algorithms (Revision 10439) and https://git.smhi.se/e8155/ece4-aogcm-oifs43/-/tree/3ee7e101.

## **Appendix A: Sequential Coupling with OASIS**


In the parallel coupling case, both OpenIFS and NEMO use their respective model time step size as the LAG parameter. For sequential coupling, the component model that computes the coupling window  $[t_n, t_{n+1}]$  first has to use its time step size minus the coupling period as a lag. In general, the component model "going first" has LAG  $\leq 0$ . The other component keeps the model time step size as the coupling lag.

We give an example of sequential atmosphere-first coupling, assuming a coupling period of  $\Delta t_{\rm cpl} = 3600\,\rm s$ , an atmosphere time step size  $\Delta t_{\rm atm} = 900\,\rm s$ , and an ocean time step size of  $\Delta t_{\rm oce} = 1800\,\rm s$ . As required by OASIS, the coupling period is an integer multiple of both model time step sizes.

Fields that are sent from OpenIFS to NEMO, e.g., the wind stresses, use LAG =  $\Delta t_{\rm atm} - \Delta t_{\rm cpl} = -2700 {\rm s}$ . Fields sent from NEMO to OpenIFS, e.g. the SST, have LAG =  $\Delta t_{\rm oce} = 1800 {\rm s}$ . Since the ocean-to-atmosphere fields have a positive LAG parameter, they are read in from a restart file in the first coupling window. In sequential ocean-first coupling, one obtains  $900 \, {\rm s}$  as the lag for atmosphere-to-ocean fields and  $-1800 \, {\rm s}$  as the lag for ocean-to-atmosphere fields.

#### 560 Appendix B: OASIS Configuration Examples for SWR

We include two excerpts from the OASIS *namcouple* configuration files for the SST coupling field, which NEMO sends to OpenIFS. We assume the same simulation parameters as in Section A and that the parallel algorithm is used in the first SWR iteration. The respective part of the *namcouple* takes the following form:

```
O_SSTSST A_SST 1 3600 2 rstos.nc EXPOUT
3 3 1 1 OC1D ASCM LAG=1800
R 0 R 0
LOCTRANS MAPPING
AVERAGE
rmp OC1D to ASCM.nc
```

These lines contain the following information (Valcke et al., 2015, §3): The coupling field is called O\_SSTSST on the source component (NEMO) and A\_SST on the target component (OpenIFS); the coupling period is  $\Delta t = 3600 \, \mathrm{s}$  and the coupling lag is equal to  $\Delta t_{\rm oce} = 1800 \, \mathrm{s}$ . Two transformations are applied by OASIS during communication: averaging over the coupling period and remapping from the  $3 \times 3$  OC1D grid to the  $1 \times 1$  ASCM grid, using the weights specified in  $rmp\_OC1D\_to\_ASCM.nc$ . Finally, the coupling fields should be sent and written out to debug files, as specified by EXPOUT. For a standard coupled run of the AOSCM, EXPORTED would also be a valid choice for the coupling field type. However, our SWR implementation requires the OASIS output files in order to reuse data from previous iterations.

For every SWR iteration after the initial guess, a different variant of the *namcouple* is used:

```
# write out current iteration
```

O\_SSTSST O\_SSTSST 1 3600 1 rstos.nc OUTPUT OC1D OC1D LAG=1800 LOCTRANS

AVERAGE



# read in from previous iteration

A SST A SST 1 3600 0 A SST.nc INPUT

The treatment of the SST is now split up into two different OASIS tasks: saving the values of the current iteration to an output file and loading data from the previous iteration. Note that here, neither regridding nor renaming coupling variables is done by OASIS, since OUTPUT and INPUT fields do not support the same transformations as when data is exchanged between components. Instead, our Python wrapper takes care of these tasks, except for computing the time average.

# Appendix C: Parameterization of albedo over sea ice in SI<sup>3</sup> and its regularized version

## C1 Standard SI<sup>3</sup> parameterization

The surface albedo  $\alpha^i$  over sea ice is a function of the ice surface temperature  $T^i$ , ice thickness  $h^i$ , snow depth  $h_s$ , and cloudiness. It is computed as a weighted average of the ice albedo below a clear sky  $(\alpha^i_{cs})$  and the albedo of ice below an overcast sky  $(\alpha^i_{os})$ , with

$$\alpha_{\rm cs}^i = P_2(\alpha_{\rm os}^i),$$

$$\alpha^i = (1 - {\rm cldf}) \times \alpha_{\rm cs}^i + {\rm cldf} \times \alpha_{\rm os}^i,$$

where  $P_2$  is a second-order polynomial function and cldf a constant parameter (typically set to cldf = 0.81). The calculation of  $\alpha^i$  thus reduces to the estimation of  $\alpha^i_{os}$ . If we set aside melt ponds (which we have turned off in our simulations),  $\alpha^i_{os}$  is computed as the sum of contributions from snow and bare ice

$$\alpha_{\text{os}}^i = (1 - \beta)\alpha_{\text{ice}}^i + \beta\alpha_{\text{snw}}^i,$$
(C1)

where it holds that  $\beta = 0$  if the snow depth  $h_s = 0$ , and  $\beta = 1$  if  $h_s > 0$ . In the following, we denote by  $\mathrm{rt}_0 = 273.15\,\mathrm{K}$  the freezing point of freshwater. The freezing cases are characterized by  $T^i < \mathrm{rt}_0$  and melting cases by  $T^i \ge \mathrm{rt}_0$ .

# C1.1 Computation of $\alpha_{\rm ice}^i$

$$h_s=0$$
 & melting case:  $\alpha_{\rm ice}^i=\alpha_{\rm mlt}^i=0.50,$  (C2a)

otherwise: 
$$\alpha_{\text{ice}}^i = \alpha_{\text{dry}}^i = 0.60.$$
 (C2b)

This value is refined depending on the ice thickness  $h_i$ :

$$0.05 < h_i \le 1.50: \qquad \alpha_{\text{ice}}^i = \alpha_{\text{ice}}^i + (0.18 - \alpha_{\text{ice}}^i) \left( \frac{\ln(1.5) - \ln(h_i)}{\ln(1.5) - \ln(0.05)} \right),$$

$$h_i \le 0.05: \qquad \alpha_{\text{ice}}^i = \alpha_{\text{oce}} + (0.18 - \alpha_{\text{oce}}) \times (20 h_i),$$

with  $\alpha_{\text{oce}} = 0.066$ . Note that the function  $\alpha_{\text{ice}}^i = F(h_i)$  is  $\mathcal{C}^0$ -continuous and is such that  $F(0) = \alpha_{\text{oce}}$ , F(0.05) = 0.18, and  $F(1.5) = \alpha_{\text{mlt}}^i$  (or  $F(1.5) = \alpha_{\text{dry}}^i$  depending on  $T^i$ ).

## C1.2 Computation of $\alpha_{\rm snw}^i$

Freezing case: 
$$\alpha_{\text{snw}}^i = \alpha_{\text{dry}}^s - (\alpha_{\text{dry}}^s - \alpha_{\text{ice}}^i) \exp(-50 \times h_s)$$
. (C3a)

Melting case: 
$$\alpha_{\text{snw}}^i = \alpha_{\text{mlt}}^s - (\alpha_{\text{mlt}}^s - \alpha_{\text{ice}}^i) \exp(-(100/3) \times h_s),$$
 (C3b)

with  $\alpha_{\mathrm{mlt}}^s = 0.75$  and  $\alpha_{\mathrm{dry}}^s = 0.85$ .

It can readily be seen that this albedo calculation exhibits discontinuities depending on the values of the input parameters  $T^i$  and  $h_s$ . This issue is even more exacerbated in the multi-category case, where the albedo is calculated as a weighted average of the albedo for each category, thereby increasing the likelihood of discontinuous behavior over time.

#### C2 Regularized version

In order to regularize the standard parameterization, we use a sigmoid function S(x) replace discontinuous jumps with a smooth transition. S(x) is defined as

$$S(x,x_0,\epsilon) = \frac{1}{1 + \exp\left(-(x-x_0)/\epsilon\right)},$$

where  $x_0$  is the central point around which the transition occurs and  $\epsilon$  defines the steepness of the transition. The objective is to replace the if-statements related to the sign of  $T^i$  and the value of  $h_s$  in the albedo computation by using this function. S(x) switches between two values 0 and 1. Roughly speaking, we have  $S(x=x_0)=1/2$ ,  $S(x=x_0+4\epsilon)\approx 1$ , and  $S(x=x_0-4\epsilon)\approx 0$ . For the switches related to  $h_s>0$  or  $h_s=0$ , we use  $\beta_{\rm snw}=S(h_s,0.01\,{\rm m},2.5\times 10^{-3}\,{\rm m})$  and for those related to the sign of  $T^i$  we use  $\beta_T=S(T^i,-0.01\,{\rm ^{\circ}C},2.5\times 10^{-3}\,{\rm ^{\circ}C})$ . Once  $\beta_{\rm snw}$  and  $\beta_T$  are computed, we can replace the condition (C2) by

$$\alpha_{\rm ice}^i = (1 - \beta_{\rm snw})\beta_T \times \alpha_{\rm mlt}^i + ((1 - \beta_{\rm snw})(1 - \beta_T) + \beta_{\rm snw}) \times \alpha_{\rm drv}^i$$

and (C3) by

$$\alpha_{\mathrm{snw}}^{i} = \left(1 - \beta_{T}\right)\left[\alpha_{\mathrm{dry}}^{s} - \left(\alpha_{\mathrm{dry}}^{s} - \alpha_{\mathrm{ice}}^{i}\right)\exp\left(-50 \times h_{s}\right)\right] + \beta_{T}\left[\alpha_{\mathrm{mlt}}^{s} - \left(\alpha_{\mathrm{mlt}}^{s} - \alpha_{\mathrm{ice}}^{i}\right)\exp\left(-(100/3) \times h_{s}\right)\right].$$

<sup>&</sup>lt;sup>9</sup>To avoid overflow problems when the surface temperature  $T^i$  takes large negative values, we use in the code the equivalent form:  $S(x, x_0, \epsilon) = 1 - 1/(1 + \exp((x - x_0)/\epsilon))$ .

Figure C1. Comparison of model output in the final time step of the 84 TOP experiments when switching the albedo parameterization.

Finally, in (C1),  $\beta$  is replaced by  $\beta_{\text{snw}}$ .


The impact on the resulting regularized albedo computation compared to the standard one is illustrated in Figure C1. As expected, the temporal evolution of the albedo is smoother, without significantly deviating from the value given by the standard parameterization. This illustrates that the regularization has not altered the physical relevance of the parameterization.

*Author contributions.* Conceptualization: VS, FL, PB, EB; Formal analysis, Software, Visualization, Investigation: VS; Funding acquisition: PB, VS; Writing – original draft preparation: VS; Writing – review & editing: FL, PB, EB.

Competing interests. The authors declare that they have no conflict of interest.

Acknowledgements. We would like to thank the AOSCM group at Stockholm University for support with model setup and data access, in particular Michail Karalis, Anna Lewinschal, Hamish Struthers, and Gunilla Svensson. Thank you furthermore to Sophie Valcke for valuable input regarding the SWR implementation and coupling algorithm switching in OASIS3-MCT. VS received financial support for this project

from The Royal Physiographic Society in Lund and the eSSENCE research program. The computations were enabled by resources provided by LUNARC, The Centre for Scientific and Technical Computing at Lund University. The research presented in this paper is a contribution to the Strategic Research Area "ModElling the Regional and Global Earth system", MERGE, funded by the Swedish government. FL and EB acknowledge funding from the French National Research Agency (ANR) under project ANR-23-CE56-0006-01 (MOTIONS).

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
