# Peer review of "Quantifying Coupling Errors in Atmosphere-Ocean-Sea Ice Models: A Study of Iterative and Non-Iterative Approaches in the EC-Earth AOSCM"

_EGUsphere, 2025_

## Referee Comment (RC1)

**Comments for:**

**Quantifying Coupling Errors in Atmosphere-Ocean-Sea Ice Models: A Study of Iterative and Non-Iterative Approaches in the EC-Earth AOSCM**

by V. Schüller *et al*.

https://doi.org/10.5194/egusphere-2025-1342

Referee : charles.pelletier@ecmwf.int

**General**

As its title suggests, the work presented in this manuscript focuses on quantifying systematic coupling errors committed when coupling ocean and atmosphere models due to the inevitable "lag" in the perceived exchange fields across their shared interface.

The authors first provide their motivations, expanding on why classical coupling methods can be qualified as "non-converged". Then, they proceed to introduce the Schwarz Waveform Relaxation (SWR) method as a framework for investigating the convergence (or lack thereof) of non-overlapping coupling problems, which the ocean – atmosphere one falls into. The authors then apply the SWR formalism to the coupled ocean—atmosphere single-column model (SCM) version of EC-Earth, a state-of-the-art Earth System Model (ESM) participating in CMIP exercises. They focus on two realistic test cases, including one involving sea-ice presence, which, to my knowledge, had not been investigated (with this lens) in the existing literature. Further investigations lead to pinpointing two specific parametrizations (the atmospheric convection scheme, and the sea-ice albedo) as potential causes of numerical instability in the coupling, which could only be identified through the SWR lenses. The authors then come to the more general conclusion that ensuring the "regular" (in the mathematical sense) character of ESM physical cores are most likely a necessary condition for ensuring the well-posedness of the underlying coupled ocean – atmosphere problem they are attempting at solving.

The manuscript is of very high quality, didactive, and contains engaging information for ESM modellers. In my opinion, it builds up on more theoretical applied mathematics literature and makes a considerable step towards applications to state-of-the-art climate models. Despite being set in a relatively idealised setting (which is perfectly understandable, considering the novel approach the authors are pursuing), the authors hint towards clear recommendations for ESM models. In that sense, this manuscript is an excellent and exciting submission for GMD.

That said, I do think that the manuscript could be made clearer, especially to the GMD readership. There also are key unaddressed methodological limitations, which I think could be at least presented and briefly discussed. I therefore am recommending this paper for publication under major revision, provided the authors address the comments

listed below. "Major" and "minor" does not mean "important" and "unimportant" comments – I think all should be accounted for, but I expect the "minor" ones to be more easily dealt with.

**Major comments**

1.  Introducing SWR and its relationship to well posedness

L. 20 – 23: I think this important part should be expanded to suit the lenses of a typical ESM (or components thereof) modeller. Before talking about SWR algorithms, it would be nice to more explicit about what current coupling methods do (or do not do), algorithmics-wise, in terms that are more intuitive to Earth system modellers.

I would say that for practical (efficiency) reasons, both models must be coupled at a frequency lower than either model's time step. and that this introduces errors (lack of feedback within one coupling window) and/or latency/staggering of the surface BC from one model to the other. And that this is unavoidable.

To be more concrete, give one example, e.g. for the parallel EC-Earth algorithm (around L. 230 – 235). At this point, I think introducing any sort of mathematical notations, or notions like "coupling window", would not really help but just saying that openIFS computes the fluxes using SSTs which are delayed compared to the model's clock, and that NEMO gets fluxes that are also late (and on top of that, these fluxes were computed from "belated" SSTs).

L. 24 – 25: I'd be slightly more explicit and say that SWR algorithms can provide "the" reference solution (if the problem is well-posed, it is "the" solution, not "a") that classical (first iteration only) methods can be compared against, thus giving an estimate of the committed error. The manuscript could also be more expansive about what SWR methods bring with respect to error quantification: without doing them, the actual proper solution to the coupled problem is not known (we don't even know whether there is one!), usually what comes out of the first iteration is just accepted as is.
There's a lot of literature of the impact of the coupling frequency on coupled model performance (e.g., Lebeaupin Brossier, 2009; Scoccimarro, 2017; Li, 2020), which I think is an illustration of what the manuscript is covering there, but maybe more familiar / graspable to the typical GMD reader. This parallel might be worth being done there.

L. 28 – 31: Drop the "typically", they're always developed independently. I have two scientific comments/questions regarding these few lines.
   a.  We don't know whether in their state-of-the-art implementations, air-sea coupling problems are well- or ill-posed, but given that they include many parametrizations that have often (always?) been developed with concerns pertaining to physical realism rather than well-posedness (and that's fine and understandable), it probably is a safe bet to consider them ill-posed until the contrary has been proven, or at least observed (like the manuscript does to some extent).

b. The manuscript says that if SWR converges, then the coupled is well-posed. Is the reciprocal true? If the answer is not known, then it's also worth being explicitly written there.

L. 257 – 262: this is an interesting paragraph. What does "obey regularity" mean? My understanding is that:
a) if the RHS is not regular enough, then the coupled problem is ill-posed.
b) a coupled problem can be ill-posed even if the RHS is regular
The manuscript is stating a). Is b) true also?
I'm not sure I understand the logics of L. 257, with the "or".
I would more pragmatical and say that since all we have is a), then we should at least strive to have regular RHS, and then elaborate about physical cores there.
But then, if b) is true (which I'm not sure of), then having regularity does not provide a guarantee of well-posedness either. It would be a necessary condition, not a sufficient one.

**2. (Non-)linear free surface, and "NEMO3.6 vs NEMO4.0"**

L. 120: In **our** 3.6 NEMO version, the volume of the oceanic column is constant... This (important) difference is known to as "nonlinear free surface" (or vertical varying layers) to the community. I think explicitly saying it might make sense. And that option was already available is NEMO3.6, so "our" NEMO3.6 would clear up potential confusions ("our" NEMO4 as well, because it's also possible to run without this option in NEMO4). I would stick to linear free surface vs nonlinear free surface (or equivalently, non-VVL vs VVL). That is what the distinction is about, the NEMO3.6 vs NEMO4 one is artificial and coming from specific physical choices.
L. 175 – 179: IMO, that part is a good example of the fundamental difference between a non-VVL and a VVL model (NEMO3.6 vs NEMO4, in your case). There's no salt being exchanged between the IFS and NEMO in either model versions. But a linear free surface does need a **virtual** salt flux, because the ocean volume is kept constant, and the prognostic variable for salinity is actual salinity (salt per unit volume). So, salinity will decrease/increases via removal/injection of water mass. On the other hand, a VVL model uses salt per unit **area** as an effective prognostic variable, and the variations in ocean volume takes care of concentrating/diluting the salt upon water mass removal/injection.
That's not the point of the manuscript, so don't be expansive on this. But I do recommend sticking to sharper terminology (linear free surface and/or (non-)VVL) whenever discussing implications on your study. The boundary condition for salinity is one.
Same applies to the heat boundary condition, for which the internal enthalpy of mass exchanges (E-P terms) only apply with the VVL. The manuscript happens to neglect them anyway, which is a reasonable assumption.

**3. One unaddressed key limitation of the SWR wrapper (from my understanding)**

L. 282 – 286: I understand the point about doing the SWR with the wrapper, etc. But there is a serious potential drawback that is never addressed: I have a strong gut feeling

that this approach, while indeed more flexible, is a lot greedier than the (comparatively intrusive) method of Marti. Marti's method will iterate over each coupling window separately and then proceed to the next window when convergence is reached. Whereas the SWR wrapper re-runs the full simulation. It's not a showstopper for your paper, especially considering it uses a 1D model for which cost does not come into question. But I still think that this point is worth being mentioned and at least mentioned. The SWR wrapper is indeed less intrusive, and that's a strength, but it's also probably much less efficient, which is a drawback.

4. Comparing coupling methods based upon "lowest error"

L. 476 – 478: I am not convinced that "coupling method that produces the lowest error" is a relevant evaluation metric. Especially considering ocean-first and atmosphere-first look like they're running a real close race. An ESM user is probably interested in a method that limits cases where the error is large. IMO, it doesn't matter to them whether the error is small, or very small, which is what the current criterium will discriminate. The user probably wants to avoid large errors, where "large" is a tolerance they are willing to accept.
This ranking criterium (lowest error) is then used as ground base for delivering one of the key conclusions of the manuscript; that ocean-first methods might be better in the presence of sea ice. But I'm not convinced that the comparison method is relevant in the first place. I think a safer conclusion (and the manuscript does say it already), is that the presence of sea ice shuffles the cards again. I would need more evidence, with other criteria, to be convinced with ocean-first as a "best candidate".
Incidentally, these conclusions arose from running one test case, in one SCM configuration. As of now, it is also not clear whether they would be robust in other configurations.

**Minor comments**

L. 40: replace "e.g." with "in our case"

L.54: for clarity, I would split this sentence in two. "...is robust in ice-free condition. This allows us to have a reference state to compare classical one-iteration coupling to, for which already after two days, temperature coupling errors can reach up to several degrees in the atmospheric boundary layer."

L. 57: Say NEMO 4.0.1 instead of "newer model versions".

L. 88: the first term of (2) is turbulence (and it's local), but the second is not, it's a nonlocal convective contribution, which I would refrain from describing as "turbulent". To me, (2) contains turbulent and convective contributions to vertical transport on subgrid scales.

L. 97: might be worth specifying that in the SCM, the Coriolis parameter is constant.

L. 103: not a huge fan of the word "local" in this context. To me, gradients are local properties of the function, because it only involves the vicinity of the point it's evaluated at (but numerical schemes assessing them are nonlocal) … I know what's meant, what I'm not sure what the relevant term should be there. Maybe just say that these terms do not involve gradients.

L. 106: "incompressible" on top of Boussinesq and hydrostatic is relevant.

L. 113: the four **large-scale liquid ocean** prognostic variables are… large-scale (or "resolved") because when used, at least TKE is also prognostic. "Liquid ocean" because the sea-ice model also has a bunch of prognostic variables.

L. 118: might be worth explicitly saying that the unlike the solar heat flux, the nonsolar heat flux does not penetrate. Which is the reason why both are treated separately.

L. 124: I wouldn't say LIM3 and SI3 are "equivalent", especially considering some of your results presented further down suggest SI3 has much better convergence properties than LIM3 (although I'm not convinced yet that it's due to the sea-ice model, it could also be the NEMO model update, and/or the VVL). "Similar" is a better fit, and it doesn't dampen your message.

L. 132: Having the bottom sea-ice temperature set to the local seawater freezing point is a (Dirichlet) boundary condition for the sea ice. It might be worth being explicitly phrased out so.

L. 144: Are there any reference/evidence to support this rather strong claim? I don't know the SCM model well, but to my understanding this statement is not true… To me:
   a. Kinetic energy should be conserved, I agree.
   b. Why isn't mass conserved with NEMO4 (and the VVL)? I think it should be…
   c. Energy is never conserved, at least due to the internal enthalpy that is never accounted for by the IFS (and that is only considered in NEMO with the VVL on). The manuscript does refer to it later in a footnote.

I'm happy to be proven wrong here, but I would suggest either proving me wrong and providing evidence for it or simply removing this slippery sentence. Incidentally, to my understanding, whether the interface is heat/momentum/mass conserving or not does not bear implications to the manuscript.

L. 171 – 174: please briefly introduce C_H (the same way it's dine for C_M above).

L. 175: well-posedness and eq. 8

L. 185: The manuscript is slightly confusing here, IMO. It is healthy for both stresses (eqs. (10a) and (10b)) to be different, because they are located at physically different interfaces (atmosphere – ice and ice – ocean, respectively). The way the manuscript is worded might make it seem like (10a) and (10b) being different is a model inconsistency. Which might not the authors' position, but the manuscript is somewhat ambiguous

there. For clarity, I would distinguish "ocean" (which, in my opinion, can include both ocean and sea ice) and "*liquid* ocean" with a clear terminology distinction.

L. 217: I understand what the authors mean with the ice concentration already being accounted for by SI3 in *$Q_i$, but I still thinks that explicitly writing $a_i * Q_i$ is more self-consistent, as $Q_i$ has been introduced as the ice – ocean heat flux in the 100% ice cover case. The model can do what it wants (and it happens to "update" the boundary condition with some sea ice induced modulation), but I think writing out $a_i * Q_i$ is accurate as well and less prone to confusion.
I also recommend doing the same for the salinity flux and write $a_i * S_t$.
And I would split (14b) into two different equations (without or with VVL). In the non-VVL case (so, NEMO3.6), are you sure that there is no (1-a_i) factor in front of the (E-P) term?

L. 229-236: I would remove the sentence starting with "At each coupling window, …", because it's covered more precisely from L.234 on. And I would move the sentence starting L. 230 ("This introduces a coupling lag") after the sentence currently ending L.236. "At each coupling time step, the coupling variables *from one model* are averaged in time over the past window to obtain the interface boundary conditions *of the other model* over the next coupling window."

L. 232: The footnote should be included as main body text there, IMO.

L. 243: ECMWF's single executable coupling could at least be mentioned there. It's sequential, atmosphere first. It's much less flexible than OASIS, so less prone to investigations and tests. But it's fast, which is why it's been done in the first place. The relevant reference is https://doi.org/10.21957/rfplwzuol

L. 267 – 268: I'd remove the last sentence -- that's just what a compiled code is.

L. 270: "Many aspects of the coupling setup **at run time** can be changed  using this file" – editing the file during runtime is probably not safe. And that's a good thing, IMO.

L. 272: "Whether an AOSCM experiment uses the parallel algorithm or one of the sequential ones **at runtime** can be controlled  by modifying the LAG parameters in namcouple."

L. 277: Is mentioning COCOA explicitly really needed? Maybe just stick to citing both papers together (Voldoire and Valcke)

L. 280: maybe "rewinding" in time?

L. 281: "basing the implementation on OASIS, with minimal changes in the ocean and atmosphere models, can facilitate its reuse in other climate models"

L. 291 – 292: This might be relevant in the conclusion paragraph as a perspective. Not here.

L. 318: Please specify the year at that point.

L. 335: and mostly, infeasible for 3D runs.

Fig. 6 and discussion: I think one remarkable point about these experimentations is that even after each coupling method differs one from another (e.g. around July $2^{nd}$ 12:00), then they can roughly reconverge (i.e., be close one to another). It's not that obvious to me – one might have expected that once the coupling error gets bad enough, then all bets are off from the next coupling window, as the model's trajectory strayed away from the reference solution. But it seems to be able to catch up. Do you have a rough idea as to why? Could it be due to the SCM framework (i.e., the top and bottom BCs essentially acting as nudging)?

Fig. 6: bit of a detail, but I think having the diurnal cycle more clearly marked via the plot grid would make the figure more readable (e.g., put emphasis on ticks at 00:00).

L. 371: Could you provide any guesses at to what is causing the hyper-sensitivity of the turbulent heat fluxes to the SST? I'm wondering whether the point at which the fluxes diverge, but the SSTs don't, is close to neutral stability. That might translate into discontinuities in the bulk formulas via the stability functions.

L. 375: I don't understand the phrasing in the parenthesis – I think there should be two separate ones?

l. 393: To be fair to the atmosphere-first method, you can say that it's indeed pretty good for the SST but has neutral performance for atmosphere fields. The current phrasing might make it seem like it's worse for atmosphere fields.

L. 405: Specify the year.

L. 496: in the presence of sea ice

L. 503 – 507: It might be outside of the scope of the manuscript, but I think this point goes beyond the coupling problem. It is about having a physical core that the dynamical core can cope with. In the physical world, sea-ice albedo can be as jumpy as it wants to, but the model numerics do not work well with sharp transitions, and we use numerical models to represent it. It can be seen as a matter of adapting (potentially doing compromises to) the physics so that the numerics work OK. Unless we work with methods that can deal with less regular functions, like Discontinuous Galerkin. But that's another topic.

---

## Referee Comment (RC3)

**Review of Schüller et al., Quantifying Coupling Errors in Atmosphere-Ocean-Sea Ice Models: A Study of Iterative and Non-Iterative Approaches in the EC-Earth AOSCM**

Schüller et al. investigate the intrinsic errors associated with a standard method of coupling atmosphere and ocean models, namely explicit parallel coupling. They do this, if my understanding is correct, by comparing a series of short single column model experiments in the Arctic and Pacific to a control run produced by a novel approach: iterating the experiment a large number of times and using coupling variables from the previous iteration to force the ensuing iteration. They also compare to serial coupling approaches in which atmosphere is run before ocean (and vice versa).

The reduction of coupling errors is a very important problem in climate and weather modelling. While Schuller et al.'s control run methodology may be not be practical for systematic use (see discussion below), their experiments produce a number of intriguing new results which may inform model development. Of particular note is the sensitivity of the coupling to the sea ice albedo parameterisation, and to a lesser extent the cloud parameterisations. This study should therefore be published. I have two major questions/concerns, which are each set out below as a discussion followed by recommendations, followed by a few more minor suggestions.

1. **The experiment design: SWR and its relationship to standard coupled model approaches**

If I have understood correctly, the authors produce (for example) a 2-day SCM run in the MOSAiC region from 0000Z 14-04-2020 to 0000Z 16-04-2020 by the following means. The coupled model is initialised from ERA5 and CMEMS and run forward for 2 days, using additional ERA5 atmospheric forcing for context where necessary, with all coupling variables saved at every coupling instance. Once the experiment ends, it is repeated from the same initial conditions, but at each coupling instance $T$, the saved variables from the coupling instance $T+1$ in the previous iteration are used as input to each model, while coupling output is saved for use in the following iteration.

(Aside: I hope I am right here about the 'jump forward' to $T+1$; without it, I could not see how the model evolution would actually change from the first iteration to the second – and for the updating process to work, the first two iterations' evolution must differ. Specifically, at iteration 2, the initial coupling exchange would otherwise just read the initial coupling variables from the first iteration, which would be identical by design; the first model timestep would then evolve in exactly the same way, and the second coupling exchange output variables would also be identical between iterations, etc.)

The authors' experiment design corresponds, as they state, to Schwarz waveform relaxation, in which an equation is repeatedly numerically integrated over a time domain and multiple space domains, with each spatial domain repeatedly updating boundary conditions. It is described as a contrast to standard explicit parallel coupling, in which each spatial domain calculates the latest boundary conditions for use by its neighbours in the ensuing coupling period.

While the SWR method is useful for the present idealised study, I think it is technically and scientifically very different to standard explicit coupling, and attempting to use it for longer experiments might be problematic, because of chaotic variations in run evolution. For iteration N, the coupling variables from iteration N-1 are used as input. This will alter the run evolution,

but it will also, presumably, impose some control on how the run evolution varies, as the atmosphere bottom boundary condition is tied closely to the previous iteration for the entire duration of the run. For the numbers of iterations considered here by the authors, this would presumably impose strict controls on how much the run evolution can vary relative to the first iteration (the standard, explicit parallel solution). For a run of sufficient length (e.g. a year) this seems unrealistic, and might indeed mean that much of the effects of improved coupling cannot be realised. An alternative outcome, too, is that the atmosphere evolution might vary chaotically from iteration to iteration such that eventually it is completely unrelated to its own lower boundary condition, the coupling variables from the previous iteration – which might cause a model error.

A more comparable approach may be the implicit solvers found throughout the atmosphere, ocean and sea ice submodels, which iterate not over the whole model period, but simply over every timestep. I am not asking the authors to carry this out, but it seems that an interesting experiment would be to carry out the SWR method coupling period by coupling period: run the parallel models forward for a coupling period and repeat, using the outputs from iteration *N* as inputs for iteration *N+1*. After a fixed number of iterations, continue to the next coupling period. I can see why such an experiment might not be useful for the present study: after the first coupling period, the model evolution would begin to vary, such that errors between standard coupling and control would no longer be easy to interpret. However, this approach seems to be scientifically more appropriate for wider use (even if still computationally very expensive).

This would maybe not matter, if it were not for the authors' L335: 'Such a large number of iterations is infeasible for long or many runs.' I think the problems of using the authors' method for long runs are more fundamental than this, and the method is better understood as a means of producing a control for these short idealised experiments, rather than as an alternative means of coupling.

**Recommendations:**

- The experiment design needs to be set out more clearly from the start, notably the fact that the iteration is performed over the whole model run (or otherwise, if I have completely misunderstood this).
- For the reasons above I think it needs to be made clear that the control run methodology is not a practical way of running a coupled climate model for longer than a few days – or if the authors disagree, justification should be given. This is not a criticism of the study; it is a highly appropriate methodology for the purpose on which the authors actually use it.
- I would like the authors to clarify in the paper how the evolution of the second iteration will differ from the first, if the coupling output from the first iteration drives coupling input for the second – is it because of a time offset?

**2. The sea ice albedo behaviour and the authors' solution**

The behaviour of the SWR method in the presence of warm air intrusions over sea ice is fascinating, and provides evidence that the coupling problem is qualitatively different where sea ice, or specifically melting sea ice, is present. Figure 10 in particular is quite shocking: the point

that in the old (LIM) sea ice model, flipping the albedo from its cold to its melting value results in surface temperature swings of 30 degrees. I would be interested to know if the authors have any idea as to a) the mechanisms, in particular negative feedbacks, behind this behaviour; b) why it is so greatly attenuated in SI[3].

The authors' albedo parameterisation test provides convincing evidence that the albedo parameterisation is responsible for most of the observed instability (presumably not all, given that the experiment still fails to converge in 3 out of 84 cases?).

On the surface, there's an interesting problem here: in the physical world, the transition in albedo between cold and melting snow/ice is presumably pretty sharp. By introducing a smooth variation, the algorithm performance might be improved despite making the model less representative of reality. In practice, the values the authors choose for epsilon are sufficiently small that this effect is probably negligible – and explainable by the fact that the model aggregates temperature over large areas that would in the real world make the transition much smoother (and in the real world the atmosphere and sea ice can react instantly to one another).

Nevertheless, it is interesting that the sigmoid parameterisation is able to resolve the oscillatory behaviour so effectively despite being so close to the original, abrupt parameterisation (due to the small values of epsilon). The ranges of Ti over which the oscillations occur in Figure 10 are sufficiently large that they could not be directly affected by the authors' sigmoid parameterisation; is this because there is a large number of variables that are 'out of balance' by the end of the iterations, driving the uncontrolled negative feedback? Presumably at some crucial point in the iterations, small oscillations must grow in the tiny region of the state space that is affected by the authors' sigmoid parameterisation?

**Recommendations:** it would be really interesting if the authors could give some ideas as to

- the mechanisms behind this oscillation
- why it is improved in SI[3]
- how the sigmoid albedo parameterisations resolve it, despite being near-identical to the abrupt parameterisations in all but a very small part of the state space

In addition, I wonder if their statement in L457-8 that the albedo discontinuity is responsible for the non-convergence needs qualifying, given that 3 experiments still fail to converge – the albedo is obviously the dominant factor, but perhaps there are other parameterisations that are still causing problems?

**3. Some additional minor comments**

- Section 3 in general: it might aid understanding to have a table of variables that are exchanged between the atmosphere, and the ice and ocean
- L251 and 261: perhaps a word of explanation here to describe the physical interpretation of 'right-hand sides' would be appropriate
- Figure 4: It is interesting that relative error stays constant for around the first 7 iterations. Could the authors comment in the paper on why this is?

- L403: recommend to reword 'it is possible to study warm air intrusions in detail'

---

## Author Comment (AC2)

**Revision Comments on**
**Quantifying Coupling Errors in Atmosphere-Ocean-Sea Ice Models:**
**A Study of Iterative and Non-Iterative Approaches in the EC-Earth AOSCM**

Valentina Schüller[a] , Florian Lemarié[b] , Philipp Birken[a] , and Eric Blayo[b]

[a] Lund University, Lund, Sweden.
[b] Univ. Grenoble Alpes, Inria, CNRS, Grenoble INP, LJK, Grenoble, France.
**Correspondence:** Valentina Schüller (valentina.schuller@math.lu.se)

**Preface**

We would like to thank both reviewers for their thorough reviews and constructive feedback. In this document, we have collected each comment, as well as our response to it. The reviewers' comments are italicized, our responses in roman. All changes are highlighted in red in the revised manuscript, listed at the end of the revised manuscript, and additionally mentioned here. Unless stated otherwise, line numbers refer to the submitted manuscript, not the revised version.

**Response to RC1 and RC2**

**Major comments**

*1. Introducing SWR and its relationship to well posedness*

*L. 20 – 23: I think this important part should be expanded to suit the lenses of a typical ESM (or components thereof) modeller. Before talking about SWR algorithms, it would be nice to more explicit about what current coupling methods do (or do not do), algorithmics-wise, in terms that are more intuitive to Earth system modellers. I would say that for practical (efficiency) reasons, both models must be coupled at a frequency lower than either model's time step. and that this introduces errors (lack of feedback within one coupling window) and/or latency/staggering of the surface BC from one model to the other. And that this is unavoidable.*
*To be more concrete, give one example, e.g. for the parallel EC-Earth algorithm (around L. 230 – 235). At this point, I think introducing any sort of mathematical notations, or notions like "coupling window", would not really help but just saying that openIFS computes the fluxes using SSTs which are delayed compared to the model's clock, and that NEMO gets fluxes that are also late (and on top of that, these fluxes were computed from "belated" SSTs).*

> We understand why the reviewer would like us to start from the state of the art and then introduce SWR; we do it this way in the abstract for this reason. However, we find the narration in the introduction clearer as it is (ESMs solve domain decomposition [DD] problems; SWR is a canonical method for DD problems; standard coupling is a version of SWR). To take into account the reviewer's comments, we have adjusted the introduction as follows:

>> Earth system models (ESMs) and general circulation models (GCMs) are large, complex computer codes coupling different submodels (components) in time and space. To this end, they exchange (boundary) data, e.g., heat fluxes and temperatures, at regular intervals. As component development progresses and resolution increases, it is expected that aspects of coupling will play a bigger role (Gross et al., 2018). We focus on atmosphere-ocean and atmosphere-ocean-sea ice coupling, where multiple sets of partial differential equations are coupled using boundary conditions. This can be seen as an example of domain decomposition without overlap.

>> Schwarz waveform relaxation (SWR) methods are iterative coupling algorithms suitable for such problems: if constructed correctly, the coupled

problem has a unique solution which the iteration converges to. Standard coupling approaches in state-of-the-art ESMs can be classified as the first iteration of an SWR algorithm. Not iterating is computationally cheap but produces a numerical coupling error at the air-sea interface. This error is separate from other numerical errors (e.g., those introduced due to non-matching grids in time and space) and from modeling errors such as those resulting from uncertainties in the parameterizations of turbulent air-sea flux components (e.g., Foken, 2006; Large, 2006).

In case of convergence, the SWR method produces a reference solution to quantify the coupling error of standard coupling algorithms in isolation. As opposed to other types of numerical convergence studies, this is possible without violating implicit assumptions of physics parameterizations on time step or grid size (Gross et al., 2018). Specifically, past studies demonstrated (…)

Regarding the specific suggestion for ll. 230–235, we have incorporated the reviewer's suggestions from the minor comments for the relevant paragraph. We think that these successfully clarify the algorithm description and that a more concrete example is not necessary.

*L. 24 – 25: I'd be slightly more explicit and say that SWR algorithms can provide "the" reference solution (if the problem is well-posed, it is "the" solution, not "a") that classical (first iteration only) methods can be compared against, thus giving an estimate of the committed error. The manuscript could also be more expansive about what SWR methods bring with respect to error quantification: without doing them, the actual proper solution to the coupled problem is not known (we don't even know whether there is one!), usually what comes out of the first iteration is just accepted as is. There's a lot of literature of the impact of the coupling frequency on coupled model performance (e.g., Lebeaupin Brossier, 2009; Scoccimarro, 2017; Li, 2020), which I think is an illustration of what the manuscript is covering there, but maybe more familiar / graspable to the typical GMD reader. This parallel might be worth being done there.*

Regarding the first aspect of your comment, please see our next response to explain why we have not changed the phrasing from "the" to "a".

We appreciate the suggestion to relate the error identified via SWR to other coupling-related errors and have incorporated it as given above.

*L. 28 – 31: Drop the "typically", they're always developed independently. I have two scientific comments/questions regarding these few lines.*

*a. We don't know whether in their state-of-the-art implementations, air-sea coupling problems are well- or ill-posed, but given that they include many parametrizations that have often (always?) been developed with concerns pertaining to physical realism rather than well-posedness (and that's fine and understandable), it probably is a safe bet to consider them ill-posed until the contrary has been proven, or at least observed (like the manuscript does to some extent).*

*b. The manuscript says that if SWR converges, then the coupled is well-posed. Is the reciprocal true? If the answer is not known, then it's also worth being explicitly written there.*

We dropped "typically," as suggested.

Regarding (a): We agree with the spirit of the comment and have chosen to incorporate it by separating the issue of non-uniqueness (a direct consequence of ill-posedness) from perturbation amplification (the gravity of which will in general vary). The former is primarily of mathematical concern, while the latter is more clearly linked to model performance. Furthermore, the latter is where we directly try to contribute with our study.

Regarding (b): We (hopefully) don't say "if SWR converges, then the coupled problem is well-posed." Instead, we say: If a coupled problem is set up correctly, SWR will converge (ll. 20-21). "Correctly" is related to properties such as well-posedness of the subproblems, the full problem, and choosing compatible interface boundary conditions. These are generally desirable properties for PDE-based numerical models.

If SWR does not converge, at least one of these properties is not fulfilled, cf. §3.2. One might get away with this sometimes, but when it reaches the point that SWR consistently does not converge, one should in our opinion investigate. The reciprocal is actually not true: Just because the iteration converges, does not mean that the underlying problem is well-posed (see your comment (a)). In that case, SWR will provide *a* reference solution, but there is no guarantee that this is the *unique* solution to the problem.

We have thus **revised** the paragraph as follows:

Since components are  developed independently, it is likely  that a given ESM solves an ill-posed problem. By this we mean that choices

made with respect to modeling and numerical algorithms result in non-unique solutions and unexpected amplification of  small perturbations . The latter aspect has direct implications on model performance, motivating the investigation of such issues. Gross et al. (2018) suggest to verify that the coupling of ESM components is formulated in a robust and consistent manner using SWR: if the iteration does not converge,  model development is  advised.

*L. 257 – 262: this is an interesting paragraph. What does "obey regularity" mean? My understanding is that:*
*a) if the RHS is not regular enough, then the coupled problem is ill-posed.*
*b) a coupled problem can be ill-posed even if the RHS is regular*
*The manuscript is stating a). Is b) true also?*
*I'm not sure I understand the logics of L. 257, with the "or".*

> **Clarification from the reviewer (RC2):** *The manuscript seems to infer that regularity is a necessary condition for the well-posedness of the coupled problem. Therefore, the phrasing "the underlying coupled problem might not "obey regularity" or could even be ill-posed" is somehow counter-intuitive to me, particularly the word "even". From my understanding of the manuscript, a coupled problem that does not obey regularity is already expected to be ill-posed, precisely because it is not regular. So why insisting on "or it could even be ill-posed", as if this consisted in a stronger statement?*

*I would more pragmatical and say that since all we have is a), then we should at least strive to have regular RHS, and then elaborate about physical cores there. But then, if b) is true (which I'm not sure of), then having regularity does not provide a guarantee of well-posedness either. It would be a necessary condition, not a sufficient one.*

> To rigorously make statement (a), one would have to study the finally resulting solution operator for an ESM, including all model equations and the full numerical discretization. This analysis is, in practice, impossible to do. That is why we are so cautious about our phrasing, stating only (as discussed above): If a coupled problem is set up "correctly", SWR will converge. If it does not converge, it cannot be set up correctly.

> Yes, (b) is in fact true: A simple example is the case of non-overlapping coupled heat equations with Dirichlet conditions: There are no regularity issues and the subproblems are well-posed, yet SWR will generally not converge.

A central contribution with our manuscript (in our opinion) is that we can pinpoint specific parameterizations that lead to clear problems (e.g., perturbation amplification). We think that it is instructive to first **show** these issues (in the numerical experiments), before elaborating more on the regularity of physics parameterizations in the introduction.

Thus, **we suggest leaving this paragraph as-is.**

*2. (Non-)linear free surface, and "NEMO3.6 vs NEMO4.0"*

*L. 120: In our 3.6 NEMO version, the volume of the oceanic column is constant... This (important) difference is known to as "nonlinear free surface" (or vertical varying layers) to the community. I think explicitly saying it might make sense. And that option was already available is NEMO3.6, so "our" NEMO3.6 would clear up potential confusions ("our" NEMO4 as well, because it's also possible to run without this option in NEMO4). I would stick to linear free surface vs nonlinear free surface (or equivalently, non-VVL vs VVL). That is what the distinction is about, the NEMO3.6 vs NEMO4 one is artificial and coming from specific physical choices.*
*L. 175 – 179: IMO, that part is a good example of the fundamental difference between a non-VVL and a VVL model (NEMO3.6 vs NEMO4, in your case). There's no salt being exchanged between the IFS and NEMO in either model versions. But a linear free surface does need a virtual salt flux, because the ocean volume is kept constant, and the prognostic variable for salinity is actual salinity (salt per unit volume). So, salinity will decrease/increases via removal/injection of water mass. On the other hand, a VVL model uses salt per unit area as an effective prognostic variable, and the variations in ocean volume takes care of concentrating/diluting the salt upon water mass removal/injection. That's not the point of the manuscript, so don't be expansive on this. But I do recommend sticking to sharper terminology (linear free surface and/or (non-)VVL) whenever discussing implications on your study. The boundary condition for salinity is one.*
*Same applies to the heat boundary condition, for which the internal enthalpy of mass exchanges (E-P terms) only apply with the VVL. The manuscript happens to neglect them anyway, which is a reasonable assumption.*

We agree that there is confusion caused by the misconception of associating a linear free surface with NEMO3.6 and a non-linear free surface with NEMO4.0, whereas both versions of the code actually have the capability to run with either option. In the revised version of the manuscript, we have made an effort to clarify this ambiguity. Beyond the ambiguity in the text, the use of different options in running these two versions of the code does not result from a deliberate decision on our part, but rather from a constraint imposed by circumstances. The development

of the SCM capability in NEMO in the context of the work by Reffray et al. (2015) was originally done under a linear free surface approximation. This was still the case in the SCM version of EC-Earth 3 (Hartung et al., 2018). Recently, the SCM mode in the development versions of EC-Earth 4 has shifted to a nonlinear free surface.

Practically, switching between the nonlinear and linear free-surface modes in NEMO 3.6 is controlled by the key_vvl cpp switch at compilation. In NEMO 4.0.1, the same functionality is managed through the namelist parameter ln_linssh. While the SCM version of EC-Earth 3 runs properly when the CPP key key_vvl is not activated, it crashes when the key is enabled. In the case of the EC-Earth 4 development version, it runs correctly when ln_linssh = F but crashes otherwise. It is regrettable that these two versions of the code cannot be run with equivalent numerical options; however, we are not in a position to address this issue in the short term.

However, as noted in the manuscript, the SWR convergence results in the absence of sea ice did not differ significantly between the model versions. Moreover, the oscillations we observed with NEMO 3.6 in the presence of sea ice were particularly pronounced in the sea ice and in atmospheric surface fluxes and temperatures (see Fig. 9 in the manuscript). We would expect that the "nonlinear vs. linear" free surface primarily impacts the atmosphere-ocean and ocean-ice interfaces, rather than the atmosphere-ice interface. See also our response to the second major comment in RC3.

> For the numerical simulations presented in Section 4, we use two versions of the EC-Earth AOSCM: EC-Earth 3 and a development version of EC-Earth 4 . In version 3 of the AOSCM, a *linear free surface* is employed, implying that the volume of the ocean column remains constant. In contrast, the AOSCM version of EC-Earth 4 uses a *nonlinear free surface*, allowing the volume to vary in response to freshwater fluxes. **[Added footnote:** In NEMO terminology, the case of a nonlinear free surface corresponds to the so-called VVL (Vertical Varying Layers) case. In recent versions of the code, the VVL terminology has been replaced by QCO (Quasi-Eulerian COordinate).**]**

**3. One unaddressed key limitation of the SWR wrapper (from my understanding)**

*L. 282 – 286: I understand the point about doing the SWR with the wrapper, etc. But there is a serious potential drawback that is never addressed: I have a strong gut feeling that this*

*approach, while indeed more flexible, is a lot greedier than the (comparatively intrusive) method of Marti. Marti's method will iterate over each coupling window separately and then proceed to the next window when convergence is reached. Whereas the SWR wrapper re-runs the full simulation. It's not a showstopper for your paper, especially considering it uses a 1D model for which cost does not come into question. But I still think that this point is worth being mentioned and at least mentioned. The SWR wrapper is indeed less intrusive, and that's a strength, but it's also probably much less efficient, which is a drawback.*

> This is a very good point, which we chose not to address in detail. We would like to emphasize that we at this point do not suggest using SWR for production runs, but for **model development**; to be clear, neither with the implementation approach by Marti et al., nor with ours. With this purpose, and since our results are clearly model-dependent, we think that ease of implementation across models is more relevant than computational cost.
>
> That being said, there are two main drawbacks to this implementation which affect the computational performance:
>
> 1. The longer the time window over which one iterates, the more iterations are necessary for SWR to terminate. In our case, we iterate over the full length of the simulation, which leads to slower and slower convergence the longer the simulation time.
> 2. There is also the additional overhead of setting up the models from scratch at each iteration (effective checkpointing, I/O costs etc.). We expect parts of this to be present in the implementation by Marti et al. (2021) as well.
>
> To properly implement SWR, one would have to address both (although the number of iterations is probably the more costly factor and should be addressed first).
>
> However, we propose that one approaches this by thinking in terms of how to change the coupler and coupling interface, instead of rewriting the time loop in the components (as done by Marti et al., 2021). We think that trying work our way "inwards", instead of beginning in the heart of the components, is a more robust foundation for such an endeavor.
>
> Since the paper is not about implementing highly performant SWR in ESMs, we have decided not to include these considerations in detail in the manuscript. However, we have added the following two sentences in the corresponding paragraph of §3.3:

The implemented solution is equivalent to SWR with piecewise constant interface data averaged over each coupling window $\Delta t_\mathrm{cpl}$, with the Schwarz window size equal to the simulation time $\mathcal{T}$. Since the number of SWR iterations necessary to converge generally grows with $\mathcal{T}$, this approach is likely unsuitable for very long simulations. However, we will see in the next section that much can be learned about coupling error sources in a given model, even when considering short time scales.

See also our response to the first major comment in RC3, which raised similar questions.

**4. Comparing coupling methods based upon "lowest error"**

*L. 476 – 478: I am not convinced that "coupling method that produces the lowest error" is a relevant evaluation metric. Especially considering ocean-first and atmosphere-first look like they're running a real close race. An ESM user is probably interested in a method that limits cases where the error is large. IMO, it doesn't matter to them whether the error is small, or very small, which is what the current criterium will discriminate.*

*The user probably wants to avoid large errors, where "large" is a tolerance they are willing to accept. This ranking criterium (lowest error) is then used as ground base for delivering one of the key conclusions of the manuscript; that ocean-first methods might be better in the presence of sea ice. But I'm not convinced that the comparison method is relevant in the first place. I think a safer conclusion (and the manuscript does say it already), is that the presence of sea ice shuffles the cards again. I would need more evidence, with other criteria, to be convinced with ocean-first as a "best candidate". Incidentally, these conclusions arose from running one test case, in one SCM configuration. As of now, it is also not clear whether they would be robust in other configurations.*

We understand and share the reviewer's criticism that the experiments, and the metric of "lowest coupling error achieved", are not sufficient to rigorously draw this conclusion. We have removed the corresponding part from the end of §4.2 and adjusted the conclusion accordingly.

We still think that it is relevant to mention the inherent difference in computing ice vs. (liquid) ocean surface variables. That is, what the comment refers to as "the presence of sea ice shuffles the cards again". One area where this becomes apparent is in the size of coupling errors for $T^i$ vs. SST. We have added two sentences on this at the end of §4.2.

We have furthermore adapted the end of §4.2, as well as the second paragraph of the conclusion, to mention that there is, at this point, no obvious strategy for obtaining low errors for atmospheric variables, neither in ice-free nor ice-covered conditions.

**Revised** end of §4.2:

The maximum errors for the SST and atmospheric humidity are very small compared to the results at the PAPA station. This is not surprising considering that (a) the ocean is almost completely covered by sea ice and thus isolated from the fast-changing atmosphere and (b) the Arctic atmosphere is very cold and dry, giving small values, variation, and errors of $q$. $T^a$ and $T^i$, on the other hand, have substantial maximum coupling errors. Note that, since the ice surface temperature is computed from an energy balance, it directly responds to changes in atmospheric surface fluxes. It is thus unsurprising that coupling errors for $T^i$ are on a similar order of magnitude as those for $T^a$, and much larger than for SST.

Finally, Figure 12 shows the binned, weighted coupling error (corresponding to Figure 8) for these four variables and the three non-iterative coupling schemes. Panels a and b show that the atmosphere-first algorithm produces the best results for ocean and sea ice variables in the EC-Earth AOSCM. A substantial amount of experiments with the other two algorithms gives large relative coupling errors (e.g., more than 25% of experiments have $e_j(T^i) \geq 0.2 e_{\max}$). As in the ice-free experiments, performance is more evenly distributed for the atmospheric variables (panels c and d). Thus, no conclusion can be drawn in terms of which coupling algorithm to pick to systematically obtain low coupling errors across all model components. ~~A closer look at the output revealed however that the ocean-first algorithm produces the lowest coupling errors in about half of the experiments (38 for humidity, 40 for atmospheric temperature), outperforming the other two algorithms. This is in contrast to the ice-free experiments, where both sequential algorithms produced similarly good results. As before, the parallel algorithm does not result in low coupling errors.~~

**Revised** Conclusion, Paragraph 2:

Experiments in ice-free conditions showed that the SWR algorithm converges consistently, allowing us to produce reference solutions and quantify the coupling error. In agreement with prior research (Marti et al., 2021), the coupling error for ocean variables is small (usually well below 0.1°C for two day simulations) and dominated by a phase error related to solar radiation. This can be mitigated by using

the atmosphere-first algorithm. The coupling error for atmospheric variables in the EC-Earth AOSCM was significantly larger in our experiments, and similarly distributed for all three non-iterative schemes. We have found a link between the magnitude of this coupling error and the convective mass flux scheme in OpenIFS, which is a discontinuous parameterization and reacts sensitively to small changes in the SST.

**Revised** Conclusion, end of the last paragraph:

For ocean and sea ice output variables, the atmosphere-first algorithm once again performs best out of the three non-iterative coupling schemes under consideration. The same cannot be said for atmospheric variables, where performance is evenly distributed. We finally note  that the fast reaction time of the sea ice component (particularly regarding ice surface temperature and albedo) makes atmosphere-sea ice coupling an inherently different problem than atmosphere-ocean coupling. Strategies to improve the coupling error of the latter (e.g., using the atmosphere-first algorithm as suggested in Marti et al., 2021) might no longer apply in presence of sea ice.

**Minor Comments**

*L. 40: replace "e.g." with "in our case"*

**Revised** as suggested.

*L.54: for clarity, I would split this sentence in two. "…is robust in ice-free condition. This allows us to have a reference state to compare classical one-iteration coupling to, for which already after two days, temperature coupling errors can reach up to several degrees in the atmospheric boundary layer."*

**Revised** as follows: We find that SWR convergence in the EC-Earth AOSCM is robust in ice-free conditions, allowing us to compute coupling errors. We find that  already after two days, temperature coupling errors in the atmospheric boundary layer can reach several degrees in magnitude.

*L. 57: Say NEMO 4.0.1 instead of "newer model versions".*

The current version states "in the newest development version," (not "newer model versions") referring to the three AOSCM versions introduced in the paragraph

above. We think it is better to clearly refer to the version of the coupled SCM here, not just the version of NEMO. **We have thus left the sentence as-is.**

*L. 88: the first term of (2) is turbulence (and it's local), but the second is not, it's a nonlocal convective contribution, which I would refrain from describing as "turbulent". To me, (2) contains turbulent and convective contributions to vertical transport on subgrid scales.*

As written in Siebesma et al. (2007), the EDMF framework is a parameterization of vertical turbulent transport, which includes a diffusive as well as a convective contribution. Our phrasing is consistent with this, which is why we **decided to leave the sentence as-is**.

*L. 97: might be worth specifying that in the SCM, the Coriolis parameter is constant.*

Note: This holds only if the AOSCM is used in Eulerian mode (for the Lagrangian AOSCM, see https://doi.org/10.5194/egusphere-2024-3709). We have thus decided to **leave this sentence as-is**.

*L. 103: not a huge fan of the word "local" in this context. To me, gradients are local properties of the function, because it only involves the vicinity of the point it's evaluated at (but numerical schemes assessing them are nonlocal) ... I know what's meant, what I'm not sure what the relevant term should be there. Maybe just say that these terms do not involve gradients.*

**Revised:** These terms  do not contain additional vertical derivatives.

*L. 106: "incompressible" on top of Boussinesq and hydrostatic is relevant.*

**Revised:** The NEMO model discretizes the oceanic primitive equations (i.e. jointly considering the Boussinesq, incompressible, and hydrostatic assumptions).

*L. 113: the four **large-scale liquid** ocean prognostic variables are... large-scale (or "resolved") because when used, at least TKE is also prognostic. "Liquid ocean" because the sea-ice model also has a bunch of prognostic variables.*

**Revised,** using "resolved" instead of "large-scale".

*L. 118: might be worth explicitly saying that the unlike the solar heat flux, the nonsolar heat flux does not penetrate. Which is the reason why both are treated separately.*

**Revised,** thanks for the suggestion! We have not changed the text here, since $Q_{ns}$ has not been introduced at this point. We have added the following sentence around l. 165 of the original manuscript, when we introduce the nonsolar heat flux: In contrast to Q_\mathrm{sr}, it does not penetrate below the ocean surface.

*L. 124: I wouldn't say LIM3 and SI3 are "equivalent", especially considering some of your results presented further down suggest SI3 has much better convergence properties than LIM3 (although I'm not convinced yet that it's due to the sea-ice model, it could also be the NEMO model update, and/or the VVL). "Similar" is a better fit, and it doesn't dampen your message.*

We did not write "similar," since we think this would raise more questions than it answers (similar in what way?). We were in contact with NEMO developers and according to them, $SI^3$ and LIM3 are equivalent in terms of the model formulation and $SI^3$ is just a different software wrapper for the same sea ice model. Unfortunately, there is no changelog documenting in more detail what differs between our version of LIM3 and our version of $SI^3$. To address the comment, we have thus decided to simply remove the second part of the sentence.

**Revised** as follows:

The differences between LIM3 and $SI^3$ are primarily at the level of the software environment, but the formulation of the sea ice model itself is equivalent.

*L. 132: Having the bottom sea-ice temperature set to the local seawater freezing point is a (Dirichlet) boundary condition for the sea ice. It might be worth being explicitly phrased out so.*

**Revised** as follows:

The boundary conditions for the heat equation follow from an energy balance at the surface and at the ice base to compute the sea ice surface temperature $T^i$ and vertical conduction fluxes, respectively. At (at the ice base the temperature is prescribed as a Dirichlet condition and assumed to be at the local freezing point).

*L. 144: Are there any reference/evidence to support this rather strong claim? I don't know the SCM model well, but to my understanding this statement is not true... To me:*
*a. Kinetic energy should be conserved, I agree.*

*b. Why isn't mass conserved with NEMO4 (and the VVL)? I think it should be...*
*c. Energy is never conserved, at least due to the internal enthalpy that is never accounted for by the IFS (and that is only considered in NEMO with the VVL on). The manuscript does refer to it later in a footnote.*
*I'm happy to be proven wrong here, but I would suggest either proving me wrong and providing evidence for it or simply removing this slippery sentence. Incidentally, to my understanding, whether the interface is heat/momentum/mass conserving or not does not bear implications to the manuscript.*

> Thanks for pointing this out. It is true that this statement might no longer hold with our version of the NEMO 4.0.1 SCM. (This also depends on whether the OpenIFS SCM conserves mass, which is not a given since, to our knowledge, pressure levels are read in from the forcing file and no continuity equation is solved). We have removed both sentences, actually:
>
> >
>
> We discuss energy conservation when introducing the boundary conditions, so it was not necessary to mention here. Regarding mass conservation, we have decided to be more explicit about the mass transport term in NEMO 4.0.1, right after Equation 9, l. 178 in the submitted manuscript:
>
> > In NEMO 4.0.1, the E-P flux is accounted for as mass transport in the free-surface evolution equation.

*L. 171 – 174: please briefly introduce C_H (the same way it's dine for C_M above).*

> **Revised** by adding: Therein, $C_{H,o}$ is the transfer coefficient for sensible heat.

*L. 175: well-posedness and eq. 8*

> ***Clarification from reviewer (RC2):*** *Sorry about this obscure comment, that's a leftover from earlier drafting that should have been removed. I was thinking about eq. 8 in itself probably having a few terms keeping the coupled problem from being well-posed. At least because $C_H$ tends to not be a regular function of the solution. Potentially also the lack of surface currents representation (although that might just be an issue for momentum fluxes, not heat ones). But it's not the point of the manuscript – please feel free to ignore this.*
>
> After clarification by the reviewer, we decided **not to address** this comment.

*L. 185: The manuscript is slightly confusing here, IMO. It is healthy for both stresses (eqs. (10a) and (10b)) to be different, because they are located at physically different interfaces (atmosphere – ice and ice – ocean, respectively). The way the manuscript is worded might make it seem like (10a) and (10b) being different is a model inconsistency. Which might not the authors' position, but the manuscript is somewhat ambiguous there. For clarity, I would distinguish "ocean" (which, in my opinion, can include both ocean and sea ice) and "liquid ocean" with a clear terminology distinction.*

> Regarding the first point, we have **revised** this part as follows:

>> We begin with kinetic energy: The  turbulent momentum flux boundary conditions are given by

> In the manuscript, "ocean" always refers to the liquid ocean and "sea ice" to sea ice. We think that this is generally quite clear from context. We agree that, in principle, it can be unclear sometimes whether "NEMO" refers to the ocean component or the ocean and sea ice components together.

> **Revised:** For this reason, we have **changed** the title of Section 2.2 "Ocean and sea ice: NEMO and SI[3] ". Furthermore, we have **added** a qualifier right before Equation 3: "The NEMO SCM equations for the ocean component are:"

*L. 217: I understand what the authors mean with the ice concentration already being accounted for by SI3 in \*Q_i\$, but I still thinks that explicitly writing $a_i * Q_i$ is more self-consistent, as $Q_i$ has been introduced as the ice – ocean heat flux in the 100% ice cover case. The model can do what it wants (and it happens to "update" the boundary condition with some sea ice induced modulation), but I think writing out $a_i * Q_i$ is accurate as well and less prone to confusion. I also recommend doing the same for the salinity flux and write $a_i * S_t$. And I would split (14b) into two different equations (without or with VVL). In the non-VVL case (so, NEMO3.6), are you sure that there is no (1-a_i) factor in front of the (E-P) term?*

> Regarding $S_t$, $Q_i$: We disagree that it is more self-consistent to multiply these quantities with $a_i$. Multiplying by $a_i$ would imply that SI[3] computes these fluxes as if the grid cell is covered by sea ice and adjusts after the fact, as is done, e.g., for the momentum fluxes. However, these quantities are computed directly (albeit implicitly) depending on sea ice volume; we explicitly mentioned this in §2.3.2. If there is very little sea ice, they will be smaller, and in ice-free conditions, they will

be 0. This is philosophically different from the approach for the other boundary conditions, which is why we chose **to leave these terms as-is.**

Regarding (14b), we decided not to split the equation, despite the reviewer's suggestion. We do it similarly in Equation (9) and want to be consistent here: We give the equation for coupling with the NEMO 3.6 SCM (the AOSCM version published in Hartung et al., 2018) and in the following text we explain what changed with the newer versions of the AOSCM. However, we have slightly adjusted the phrasing to be clearer about this term vanishing in case of the free surface:

> while in NEMO 4.0.1 the contribution from $E_o$-P vanishes and instead appears in is applied to the free-surface evolution equation.

Regarding the (1-a_i)-factor in front of E-P: Again, this weighting is not actually applied in the code, so it would be misleading to write it in the manuscript. However, we fixed the evaporation flux to only denote the evaporation over ocean $E_o$, see also the adjusted caption of Figure 1 and the motivation for this in our list of additional changes.

*L. 229-236: I would remove the sentence starting with "At each coupling window, …", because it's covered more precisely from L.234 on. And I would move the sentence starting L. 230 ("This introduces a coupling lag") after the sentence currently ending L.236. "At each coupling time step, the coupling variables from one model are averaged in time over the past window to obtain the interface boundary conditions of the other model over the next coupling window."*

> **Revised** as suggested.

*L. 232: The footnote should be included as main body text there, IMO.*

> **Revised** as suggested.

*L. 243: ECMWF's single executable coupling could at least be mentioned there. It's sequential, atmosphere first. It's much less flexible than OASIS, so less prone to investigations and tests. But it's fast, which is why it's been done in the first place. The relevant reference is https://doi.org/10.21957/rfplwzuol*

> **Revised as follows:**

> It is also possible to use a sequential coupling algorithm, as done, e.g., by the ECMWF (Mogensen et al., 2012). As the name suggests, the submodels are run after each other in this configuration. Two variants exist, the

sequential atmosphere-first and the sequential ocean-first algorithms, the former depicted in Figure 2b. In practice, the sequential ocean-first algorithm is not used by state-of-the-art general circulation models (Marti et al., 2021).

*L. 267 – 268: I'd remove the last sentence -- that's just what a compiled code is.*

**Revised** as suggested.

*L. 270: "Many aspects of the coupling setup **at run time** can be changed  using this file" – editing the file during runtime is probably not safe. And that's a good thing, IMO.*

**Revised** by removing "at runtime" here completely to avoid confusion.

*L. 272: "Whether an AOSCM experiment uses the parallel algorithm or one of the sequential ones at runtime can be controlled at runtime by modifying the LAG parameters in namcouple."*

**Revised** as suggested.

*L. 277: Is mentioning COCOA explicitly really needed? Maybe just stick to citing both papers together (Voldoire and Valcke)*

**Revised** as suggested.

*L. 280: maybe "rewinding" in time?*

**Revised** as suggested.

*L. 281: "basing the implementation on OASIS, with minimal changes in the ocean and atmosphere models, can facilitate its reuse in other climate models"*

**Addressed in conjunction with the following comment.**

*L. 291 – 292: This might be relevant in the conclusion paragraph as a perspective. Not here.*

We think this point is relevant for the implementation section; it explains potential extensions of the classical SWR algorithm and links this work to recent developments in other multiphysics coupling literature. Putting it in the conclusion would put too much emphasis on this aspect, since we at this point do not recommend implementing advanced SWR variants in climate codes. However, it is true that it is out of place in the current paragraph.

**Revised:** We reworded the sentence, moved it to the first paragraph in §3.2, and addressed the comment on L. 281. It now ends as follows:

This is in contrast to the implementation of Marti et al. (2021), where the main time loops of the atmosphere and ocean models were adjusted significantly to support "rewinding " in time. Designing the implementation based on the coupling software, with minimal changes in the ocean and atmosphere models, allows to naturally extend the basic SWR algorithm, e.g., by black-box acceleration methods such as a relaxation step or Quasi-Newton approaches (Rüth et al., 2021). More importantly, it is easier to reuse in other models, since OASIS is widely used in the climate community (Craig et al., 2017).

*L. 318: Please specify the year at that point.*

> **Clarification from reviewer (RC2):** *Sorry, wrong line numbering. The year is indeed there L. 318. I would just also repeat the year anytime a date is mentioned, e.g. L. 321 and L. 405.*

> **Revised.** We added the years in l. 321 and l. 405, as suggested. For readability reasons, we did not add the year "anytime" we mention a date, but only when introducing the dates used in a given experiment. We also switched to a more consistent date formatting (12 April 2020, 00:00 UTC, i.e., DD Month YYYY, HH:MM).

*L. 335: and mostly, infeasible for 3D runs.*

> **Revised:**

> > Such a large number of iterations is infeasible for long or many runs, especially in 3D.

*Fig. 6 and discussion: I think one remarkable point about these experimentations is that even after each coupling method differs one from another (e.g. around July 2nd 12:00), then they can roughly reconverge (i.e., be close one to another). It's not that obvious to me – one might have expected that once the coupling error gets bad enough, then all bets are off from the next coupling window, as the model's trajectory strayed away from the reference solution. But it seems to be able to catch up. Do you have a rough idea as to why? Could it be due to the SCM framework (i.e., the top and bottom BCs essentially acting as nudging)?*

> This is a good observation and we do think this point is specific to the coupled AOSCM. We did an experiment to investigate how perturbations in the initial conditions would evolve in the AOSCM depending on the coupling scheme (in icefree conditions); this is documented and discussed on pp. 46–47 and 54 in the MSc thesis by Schüller (2023). From this we concluded that the large-scale forcing in the OpenIFS SCM throughout the column, not the top and bottom BCs, constrains the evolution of the AOSCM significantly.

*Fig. 6: bit of a detail, but I think having the diurnal cycle more clearly marked via the plot grid would make the figure more readable (e.g., put emphasis on ticks at 00:00).*

**Revised** as suggested: We made the corresponding labels bold and increased the grid opacity at 00:00.

*L. 371: Could you provide any guesses at to what is causing the hyper-sensitivity of the turbulent heat fluxes to the SST? I'm wondering whether the point at which the fluxes diverge, but the SSTs don't, is close to neutral stability. That might translate into discontinuities in the bulk formulas via the stability functions.*

Indeed, the sensitivity of the heat fluxes to the SST seems to be related to discontinuities in the mass flux scheme. We have not added this plot in the manuscript, but Fig. 5.3a in the MSc thesis by Schüller (2023), reproduced below, shows that the fluxes start diverging when the coupling algorithms start giving different boundary layer types. Note that the difference here is not between stable vs. unstable conditions, but rather between different (unstable) boundary layer types. It was because of this observation that we tried turning off the mass flux scheme, resulting in Figure 7 in our manuscript.

[Figure]

*L. 375: I don't understand the phrasing in the parenthesis – I think there should be two separate ones?*

Yes, there was a typo, thanks for pointing it out!

**Revised version:** Indeed, the vertical temperature profiles differ strongly after July 2, while turning off the mass flux scheme (namelist parameter LECUMF) yields a very small coupling error, cf. Figure 7).

*l. 393: To be fair to the atmosphere-first method, you can say that it's indeed pretty good for the SST but has neutral performance for atmosphere fields. The current phrasing might make it seem like it's worse for atmosphere fields.*

We do not think that the current phrasing makes it seem that the atmosphere-first scheme behaves worse than the other methods for atmosphere fields here and have **left the current phrasing as-is**. However, we think our changes in response to Major Comment 4 address this aspect and make it clear that in general, all three non-iterative schemes behave similarly for atmospheric fields. We hope the reviewer agrees.

*L. 405: Specify the year.*

**Revised** as suggested.

*L. 496: in the presence of sea ice*

**Revised** as suggested.

*L. 503 – 507: It might be outside of the scope of the manuscript, but I think this point goes beyond the coupling problem. It is about having a physical core that the dynamical core can cope with. In the physical world, sea-ice albedo can be as jumpy as it wants to, but the model numerics do not work well with sharp transitions, and we use numerical models to represent it. It can be seen as a matter of adapting (potentially doing compromises to) the physics so that the numerics work OK. Unless we work with methods that can deal with less regular functions, like Discontinuous Galerkin. But that's another topic.*

We agree with the reviewer that it is vital for effective ESM development to consider jointly the assumptions required by different model components in terms of regularity, resolution, etc. Ensuring proper physics-dynamics coupling will certainly continue to play a bigger role and we think our manuscript does contribute to this area. However, we have decided not to add a comment of this nature in the conclusion, since we consider it out-of-scope for this article.

**Response to RC3**

**Major Comments**

*1. The experiment design: SWR and its relationship to standard coupled model approaches*

*If I have understood correctly, the authors produce (for example) a 2-day SCM run in the MOSAiC region from 0000Z 14-04-2020 to 0000Z 16-04-2020 by the following means. The coupled model is initialised from ERA5 and CMEMS and run forward for 2 days, using additional ERA5 atmospheric forcing for context where necessary, with all coupling variables saved at every coupling instance. Once the experiment ends, it is repeated from the same initial conditions, but at each coupling instance T, the saved variables from the coupling instance T+1 in the previous iteration are used as input to each model, while coupling output is saved for use in the following iteration.*

*(Aside: I hope I am right here about the 'jump forward' to T+1; without it, I could not see how the model evolution would actually change from the first iteration to the second – and for the updating process to work, the first two iterations' evolution must differ. Specifically, at iteration 2, the initial coupling exchange would otherwise just read the initial coupling variables from the first iteration, which would be identical by design; the first model timestep would then evolve in exactly the same way, and the second coupling exchange output variables would also be identical between iterations, etc.)*

*The authors' experiment design corresponds, as they state, to Schwarz waveform relaxation, in which an equation is repeatedly numerically integrated over a time domain and multiple space domains, with each spatial domain repeatedly updating boundary conditions. It is described as a contrast to standard explicit parallel coupling, in which each spatial domain calculates the latest boundary conditions for use by its neighbours in the ensuing coupling period.*

*While the SWR method is useful for the present idealised study, I think it is technically and scientifically very different to standard explicit coupling, and attempting to use it for longer experiments might be problematic, because of chaotic variations in run evolution. For iteration N, the coupling variables from iteration N-1 are used as input. This will alter the run evolution, but it will also, presumably, impose some control on how the run evolution varies, as the atmosphere bottom boundary condition is tied closely to the previous iteration for the entire duration of the run. For the numbers of iterations considered here by the authors, this would presumably impose strict controls on how much the run evolution can vary relative to the first iteration (the standard, explicit parallel solution). For a run of sufficient length (e.g. a year) this seems unrealistic, and might indeed mean that much of the effects of improved coupling cannot be realised. An alternative outcome, too, is that*

*the atmosphere evolution might vary chaotically from iteration to iteration such that eventually it is completely unrelated to its own lower boundary condition, the coupling variables from the previous iteration – which might cause a model error.*

*A more comparable approach may be the implicit solvers found throughout the atmosphere, ocean and sea ice submodels, which iterate not over the whole model period, but simply over every timestep. I am not asking the authors to carry this out, but it seems that an interesting experiment would be to carry out the SWR method coupling period by coupling period: run the parallel models forward for a coupling period and repeat, using the outputs from iteration N as inputs for iteration N+1. After a fixed number of iterations, continue to the next coupling period. I can see why such an experiment might not be useful for the present study: after the first coupling period, the model evolution would begin to vary, such that errors between standard coupling and control would no longer be easy to interpret. However, this approach seems to be scientifically more appropriate for wider use (even if still computationally very expensive).*

*This would maybe not matter, if it were not for the authors' L335: 'Such a large number of iterations is infeasible for long or many runs.' I think the problems of using the authors' method for long runs are more fundamental than this, and the method is better understood as a means of producing a control for these short idealised experiments, rather than as an alternative means of coupling.*

***Recommendations:***

- *The experiment design needs to be set out more clearly from the start, notably the fact that the iteration is performed over the whole model run (or otherwise, if I have completely misunderstood this).*
- *For the reasons above I think it needs to be made clear that the control run methodology is not a practical way of running a coupled climate model for longer than a few days – or if the authors disagree, justification should be given. This is not a criticism of the study; it is a highly appropriate methodology for the purpose on which the authors actually use it.*
- *I would like the authors to clarify in the paper how the evolution of the second iteration will differ from the first, if the coupling output from the first iteration drives coupling input for the second – is it because of a time offset?*

**Response:**

Before addressing this comment in detail, we would like to mention that this comment touches upon aspects similar to the third major comment in RC1. Please see our response above to complement the following response.

The interpretation with the "jump forward" is correct. This time shift in the coupling data is part of the postprocess_iteration(k) step in Algorithm 1 and explained in the technical report by Valcke (2021) referenced in the paper.

Indeed, the converged SWR solution is similar to using an implicit time-stepping method for the coupling terms; note that these coupling algorithms are therefore sometimes also called "implicit," as opposed to the standard, explicit algorithms. The solution is obtained by a fixed-point iteration. Whether the iteration converges, and how quickly, is related to properties of the model equations (e.g., Lipschitz continuity of the right-hand sides), and to the length of the interval over which one iterates.

It is true that we are iterating over the whole simulation; this can be seen in Algorithm 1, Figure 3, and is explained in ll. 287–291 in the submitted manuscript. Furthermore, we hope that the first sentence we added as a response to RC1, Major Comment 3, makes this aspect even more clear, thus **addressing the first recommendation:**

> Since the number of SWR iterations necessary to converge generally grows with $\mathcal{T}$, this approach is likely unsuitable for very long simulations.

Since the number of iterations will grow with longer simulations, and since we iterate over the length of the simulation, this approach is indeed limited to short simulations. We think that second sentence we added as a response to RC1, Major Comment 3, **addresses the second recommendation** in RC3:

> However, we will see in the next section that much can be learned about coupling error sources in a given model, even when considering short time scales.

A note on the second-to-last paragraph in the reviewer's discussion *("A more comparable approach may be…")*: The reviewer's suggestion to iterate over every coupling window was done in 3D by Marti et al. (2021) and is a standard way of implementing implicit coupling/SWR in coupling software. In case SWR converges, both implementation variants give the same solution, up to perturbations stemming from floating-point precision: The output of a component at time t depends only on previous time steps, prescribed forcing, and interface data. Thus, if the coupling window sizes are equal, $[t_0, t_1]$ produces the exact same iterations, no matter the size of the Schwarz window. The converged result of $[t_0, t_1]$ propagates to later time

steps, which is equivalent to first iterating over [$t_0$, $t_1$] until convergence, then continuing with [$t_1$, $t_2$], etc.

To address the **third recommendation**, we have added two sentences at the beginning of Section 3.2 and a new figure to illustrate the elimination of the coupling lag in the converged limit of SWR (Figure 3b).

> In iteration $k$ and coupling window $[t_{n}, t_{n+1}]$, a component reads coupling data from the same coupling window but the previous iteration, $k-1$. In case SWR converges, one obtains a solution without the coupling lag in the limit $k\to\infty$, cf. Figure 3b.

*2. The sea ice albedo behaviour and the authors' solution*

*The behaviour of the SWR method in the presence of warm air intrusions over sea ice is fascinating, and provides evidence that the coupling problem is qualitatively different where sea ice, or specifically melting sea ice, is present. Figure 10 in particular is quite shocking: the point that in the old (LIM) sea ice model, flipping the albedo from its cold to its melting value results in surface temperature swings of 30 degrees. I would be interested to know if the authors have any idea as to a) the mechanisms, in particular negative feedbacks, behind this behaviour; b) why it is so greatly attenuated in SI3.*

*The authors' albedo parameterisation test provides convincing evidence that the albedo parameterisation is responsible for most of the observed instability (presumably not all, given that the experiment still fails to converge in 3 out of 84 cases?).*

*On the surface, there's an interesting problem here: in the physical world, the transition in albedo between cold and melting snow/ice is presumably pretty sharp. By introducing a smooth variation, the algorithm performance might be improved despite making the model less representative of reality. In practice, the values the authors choose for epsilon are sufficiently small that this effect is probably negligible – and explainable by the fact that the model aggregates temperature over large areas that would in the real world make the transition much smoother (and in the real world the atmosphere and sea ice can react instantly to one another).*

*Nevertheless, it is interesting that the sigmoid parameterisation is able to resolve the oscillatory behaviour so effectively despite being so close to the original, abrupt parameterisation (due to the small values of epsilon). The ranges of Ti over which the oscillations occur in Figure 10 are sufficiently large that they could not be directly affected by the authors' sigmoid parameterisation; is this because there is a large number of variables that are 'out of balance' by the end of the iterations, driving the uncontrolled*

*negative feedback? Presumably at some crucial point in the iterations, small oscillations must grow in the tiny region of the state space that is affected by the authors' sigmoid parameterisation?*

***Recommendations:***

*it would be really interesting if the authors could give some ideas as to*

- *the mechanisms behind this oscillation*
- *why it is improved in SI3*
- *how the sigmoid albedo parameterisations resolve it, despite being near-identical to the abrupt parameterisations in all but a very small part of the state space*

*In addition, I wonder if their statement in L457-8 that the albedo discontinuity is responsible for the non-convergence needs qualifying, given that 3 experiments still fail to converge – the albedo is obviously the dominant factor, but perhaps there are other parameterisations that are still causing problems?*

**Response:**

**On the oscillations in NEMO 3.6/LIM3.** First of all, we would like to emphasize that the oscillations we observed with the EC-Earth 3 AOSCM are **not** due to the melting/drying albedo parameterization. They occur whenever sea ice is present, no matter if the surface temperature is below or equal to 0°C. That is, even when SWR converges with the standard albedo parameterization in the new ice model SI3, large-amplitude oscillations form with LIM3; see the first row of Figure 9. We have also verified that modifying the albedo parameterization in LIM3 has no effect on the oscillations, even when returning a constant albedo (Footnote 8, described in detail in the code repository).

We were not able to identify the root cause of the differences between SI3 and LIM3. Unfortunately, there is no documentation describing what exactly changed between the two ice models, especially in the SCM variants. Besides the albedo parameterization, we **know** about the following changes:

1. the initial files for SI3 contain fewer variables and more of the initialization happens internally;
2. the NEMO 4.0.1 SCM uses a free ocean surface, which will affect the ocean boundary conditions below the atmosphere *and* below ice;
3. since changes in SI3 are mostly on a software engineering level, in particular the interface to reading in atmospheric fluxes has changed. One would have to go

through the related modules in detail to check whether any term is read in differently than before;

We think that initialization is an unlikely cause for the problem, since the AOSCM with NEMO 3.6/LIM3 produces reasonable results in the first SWR iteration/with standard coupling schemes (Figure 9 in the manuscript). Since the oscillations are so clearly visible in ice surface variables, we furthermore think that the ice-ocean interface is an unlikely source of the oscillations in LIM3. This leaves us with the last option, which introduces a lot of work but in the end does not clearly contribute to model development: the large-amplitude oscillations are no longer a problem in SI3 and they only seemed to affect SWR, which is not a method used in production. We thus decided not to continue our investigation at this point. We furthermore decided not to include this detailed discussion in the manuscript, since, at this point, there is no clear takeaway from it.

**On the oscillations in NEMO 4.0.1/SI3.** The change in albedo at the sign change of T_I,s from nonnegative to negative is discontinuous. Notably, temperature changes that are numerically zero (e.g., from $10^{-15}$°C to $-10^{-15}$°C) can result in an albedo jump on the order of $10^{-1}$ from one iteration (or one coupling window) to the next. The albedo jump causes changes in atmospheric heat fluxes, altering the surface energy balance in the next iteration (or coupling window), once again potentially triggering a jump in albedo values. If the equilibrium temperature is very close to 0°C, these back-and-forth jumps can occur again and again, manifesting as an oscillating iteration. The main cause for the oscillations here is that the change in albedo is **independent** of the change in surface temperature/heat fluxes, amplifying arbitrarily small changes in the equilibrium temperature up to the order of the jump, $10^{-1}$. By allowing the albedo to change continuously at 0°C, we avoid this amplification, allowing the fixed-point iteration to stabilize at an "intermediate" albedo between the melting and drying one. We hope this elaboration clarifies the reviewer's questions.

We have **revised** the sentence in question as follows, following the reviewer's recommendation:

> Thus we can conclude that the discontinuity at $T^i=0$°C in the SI$^3$ albedo computation is responsible for the consistent non-convergence of the SWR algorithm during the warm air intrusions.

**Minor Comments**

*Section 3 in general: it might aid understanding to have a table of variables that are exchanged between the atmosphere, and the ice and ocean*

> This overview is given in Figure 1 and referenced in Section 3 at multiple points. We have furthermore updated this figure to be more consistent with the text in Section 2 (see the additional changes below).

*L251 and 261: perhaps a word of explanation here to describe the physical interpretation of 'right-hand sides' would be appropriate*

> We have **revised** this by referring back to the PDEs given in Section 2.1 and 2.2:

> > SWR converges under certain conditions on the well-posedness of the underlying coupled problem, in particular Lipschitz-continuity of right-hand sides (Janssen and Vandewalle, 1996), here given in Equations (2) and (3), and a correct choice of interface boundary conditions (Gander, 2006).

*Figure 4: It is interesting that relative error stays constant for around the first 7 iterations. Could the authors comment in the paper on why this is?*

> Thank you, it is true that we did not comment on this aspect at all. We have **added** the following two sentences at the beginning of the paragraph, referencing useful literature on SWR convergence:

> > The relative error stays near-constant for about seven iterations, before decaying roughly linearly every other iteration. Such behavior is expected for parallel SWR with long Schwarz windows (e.g., Janssen and Vandewalle, 1996; Gander, 2006). The figure shows (...)

*L403: recommend to reword 'it is possible to study warm air intrusions in detail'*

> Revised as follows:

> For this reason, warm air intrusions are suitable to be studied with  a coupled SCM, informing both physical understanding and model development.

**Additional Changes to the Manuscript**

- Minor editorial changes (e.g., superfluous parentheses).
- We found a typo in the code when computing the maximum coupling errors for atmospheric temperatures. We have updated the code, cited the new release, and updated the numbers: 3.99°C instead of 3.66°C for the ice-free experiments at the PAPA station (Eq. 18), and 7.48°C instead of 3.96°C for the ice-covered YOPP TOP experiments (Eq. 19). This does not affect the conclusions we draw in the text.
- We adjusted the notation in Figure 1 to be fully consistent with the notation introduced in Section 2.3. We also extended the caption of Figure 1 to mention some technical details that we omitted from the text. We now also use the previously introduced Q_ns,o in Equation (14a), instead of repeating the individual fluxes contributing to this term.
- We added a new panel to Figure 12 to illustrate the magnitude of the differences between standard coupling schemes and the SWR reference solution in case of sea ice. There, we show the vertical temperature profile in the atmosphere at the end of the two-day simulation that produced the largest coupling error. In the text, we have added two sentences on this figure:

   In Figure 12a, we show the atmospheric temperature profile for the experiment where $e(T^a)$ is maximal (17 April 2020, 16:00 UTC); in this case, the atmosphere-first algorithm produces the worst result. As observed in Figure 7, the magnitude of the error is related to the strong sensitivity of the boundary layer parameterizations to surface variables.

---

## Referee Report (RR1)

**Comments for version 2 of:**

**Quantifying Coupling Errors in Atmosphere-Ocean-Sea Ice Models: A Study of Iterative and Non-Iterative Approaches in the EC-Earth AOSCM**

by V. Schüller et al.

https://doi.org/10.5194/egusphere-2025-1342

Referee: charles.pelletier@ecmwf.int

As stated in my first report, the manuscript was already strong to start with. I think this revised version gained in clarity, thus making it even stronger. The authors' response to my and the other reviewer's comments and the related updates (or lack thereof, with justifications) are convincing.

That said, there still are a few minor points on which the manuscript could be made clearer than the current revised version. The authors' response to reviewers' comments contains valuable information that would be worth being more explicitly worded in the main body text. Therefore, I recommend publication in GMD, provided the following two minor comments (plus a typo) are addressed.

I appreciate and respect that the authors' opinions may differ from mine, but I honestly think that accounting for these last small remarks would benefit the manuscript and eventually make it an even better GMD submission.

Line numbers refer to the manuscript's version 2.

**On introducing the SWR, well-posedness etc.**

- L. 21 22: I would slightly rephrase to "if the coupled problem is well-posed, then it has a unique solution which the iteration converges to", which explicitly refers to "well-posedness", as this is a key. The initial phrasing could also have been misunderstood: the "correctly constructed" thing could have either been the coupled problem itself (which is what the authors meant), or the SWR algorithm it tries to solve.
- Line 36, I suggest adding one further sentence for insisting on the utility (and limitations) of the SWR: while SWR cannot formally prove whether a coupled problem is well-posed, it does provide a valuable stress-test on the robustness of the model formulation's robustness. This might sound like a repetition of the phrase L. 34 35, especially to domain experts like the authors, but I think this point is important enough and might be subtle to grasp for some of the GMD readership, so insisting might be worthwhile. And it is in line with one of the main messages of the manuscript, on the sea-ice albedo and atmospheric convection irregularity.

**On the mixed ocean/ice heat boundary conditions**

I still do not agree with the authors' choice not to explicitly write \$a\_i\$ in Eq. 14, but I could live with it, provided the authors add one sentence explicitly saying that \$Q\_i\$

already accounts for \$a\_i\$. I think that point is relevant, even if the scaling is done implicitly by SI3 (and rightfully so). In their answer to my initial concern, the authors say that \$2.3.2 "explicitly mentioned" this, but:

- Where is that explicit mention in §2.3.2?
- \$2.3.2 is not a relevant location for this anyway, because it treats the 100% ice cover case, for which \$a\_i\$ does not bear much meaning. Right after Eq. 14 is a better spot, IMO.

L. 244: "the" boundary condition (not "The")

---

## Author Response (AR2)

**Revision Comments on Version 2 of Quantifying Coupling Errors in Atmosphere-Ocean-Sea Ice Models: A Study of Iterative and Non-Iterative Approaches in the EC-Earth AOSCM**

Valentina Schüllera , Florian Lemariéb , Philipp Birkena , and Eric Blayob

Correspondence: Valentina Schüller (valentina.schuller@math.lu.se)

| Preface                                           | 1 |
|---------------------------------------------------|---|
| Response to Minor Comments from Referee Report #1 | 2 |
| On introducing the SWR, well-posedness etc.       | 2 |
| On the mixed ocean/ice heat boundary conditions   | 2 |
| Further Changes to the Manuscript                 | 3 |

**Preface**

We thank the reviewers for their reports and in particular reviewer #1 for the additional comments on the manuscript. We have addressed them as presented in the following pages. In the revised draft, they are marked in blue.

<sup>a Lund University, Lund, Sweden.

<sup>b Univ. Grenoble Alpes, Inria, CNRS, Grenoble INP, LJK, Grenoble, France.

**Response to Minor Comments from Referee Report #1**

**On introducing the SWR, well-posedness etc.**

L. 21 – 22: I would slightly rephrase to "if the coupled problem is well-posed, then it has a unique solution which the iteration converges to", which explicitly refers to "well-posedness", as this is a key. The initial phrasing could also have been misunderstood: the "correctly constructed" thing could have either been the coupled problem itself (which is what the authors meant), or the SWR algorithm it tries to solve.

**Revised as suggested.**

Line 36, I suggest adding one further sentence for insisting on the utility (and limitations) of the SWR: while SWR cannot formally prove whether a coupled problem is well-posed, it does provide a valuable stress-test on the robustness of the model formulation's robustness. This might sound like a repetition of the phrase L. 34 – 35, especially to domain experts like the authors, but I think this point is important enough and might be subtle to grasp for some of the GMD readership, so insisting might be worthwhile. And it is in line with one of the main messages of the manuscript, on the sea-ice albedo and atmospheric convection irregularity.

**Revised as follows:**

if the iteration does not converge, model development is advised. That is, one uses SWR not to formally obtain well-posedness results, but as a numerical stress test that specifically addresses the coupling layer.

**On the mixed ocean/ice heat boundary conditions**

I still do not agree with the authors' choice not to explicitly write \$a\_i\$ in Eq. 14, but I could live with it, provided the authors add one sentence explicitly saying that \$Q\_i\$ already accounts for \$a\_i\$. I think that point is relevant, even if the scaling is done implicitly by SI3 (and rightfully so). In their answer to my initial concern, the authors say that \$2.3.2 "explicitly mentioned" this, but:

- Where is that explicit mention in §2.3.2?
- \$2.3.2 is not a relevant location for this anyway, because it treats the 100% ice cover case, for which \$a\_i\$ does not bear much meaning. Right after Eq. 14 is a better spot, IMO.

We think there was a typo in our last response, we regret the error. In the revised manuscript in 2.3.3, right after Eq. 14, we explicitly wrote "SI³ takes into account the ice area fraction in computing \(Q^i\) and \(\mathcal{S}\_t\)." To make this clearer, we have **revised** this by explicitly repeating the variable as well:

SI3 takes into account the ice area fraction \$a\_i\$ in computing \$Q^i\$ and \$\mathcal{S}\_t\$.

L. 244: "the" boundary condition (not "The")

Revised as suggested.

**Further Changes to the Manuscript**

1. Reviewer #1 had an additional comment on the manuscript that was not part of this review report but communicated personally:

Eq. (8b), I think \$c\_p^a\$ should be substituted with \$\Lambda^a\$; it's the specific enthalpy of fusion that should be there, I believe. Unless I'm mistaken, you need this for (8b) to have the correct dimension / units.

There were two mistakes in Eq. (8b) and, likewise, Eq. (11b), which we have fixed in accordance with the ECMWF's IFS documentation by

- 1. replacing specific heat c\_p^a with latent heat of fusion L\_q,
- 2. replacing the transfer coefficient C H with a transfer coefficient C Q.

We have furthermore adapted the sentence following Eq. 8b as follows:

Therein,  $C_{H,o}$  and  $C_{Q,o}$  are the transfer coefficients for sensible and latent heat, respectively, and  $L_{Q}$  denotes the latent heat of fusion.

2. Finally, we have updated the acknowledgements as follows:

Thank you furthermore to Sophie Valcke for valuable input regarding the SWR implementation and coupling algorithm switching in OASIS3-MCT, as well as Charles Pelletier and an anonymous reviewer for valuable comments on the manuscript.